# Updating Parametric Knowledge with Context Distillation Retains Post-Training Capabilities

**Shankar Padmanabhan** [1]  **Mustafa Omer Gul** [1]  **Tanya Goyal** [1]

## Abstract

Post-training endows pretrained LLMs with a variety of desirable skills, including instruction-following, reasoning, and others. However, these post-trained LLMs only encode knowledge up to a cut-off date, necessitating continual adaptation. Unfortunately, existing solutions cannot simultaneously learn new knowledge from an adaptation document corpora and mitigate the forgetting of earlier learned capabilities. To address this, we introduce Distillation via Split Contexts (DiSC), a simple context-distillation based approach for continual knowledge adaptation. DiSC derives student and teacher distributions by conditioning on distinct segments of the training example and minimizes the KL divergence between the shared tokens. This allows us to efficiently apply context-distillation without requiring explicit generation steps during training. We run experiments on four post-trained models and two adaptation domains. Compared to prior finetuning and distillation methods for continual adaptation, DiSC consistently reports the best trade-off between learning new knowledge and mitigating forgetting of previously learned skills like instruction-following, reasoning, and factual knowledge.[2]

## 1. Introduction

Frontier large language models (LLMs) owe much of their general-purpose problem solving abilities, such as reasoning, coding, and instruction-following, to extensive post-training and alignment procedures. However, these LLMs remain static reflections of the data they were exposed to in their training phase while many real-world applications require adapting these LLMs after initial deployment.

We can broadly classify continual adaptation goals into two categories: adapting a model's *capabilities* (e.g., to learn a new task or modify behavior), and learning new *knowledge* (e.g., data from after pretraining, internal company information, etc.). Although standard finetuning for these new goals often results in severe catastrophic forgetting in both settings, recent work has shown promising results for capability adaptation when the task domains are amenable to RL-training. In particular, Shenfeld et al. (2025) and Chen et al. (2025) show that RL-based updates can mitigate catastrophic forgetting as they implicitly favor solutions that minimize the KL divergence from the initial policy.

However, continually adapting models to new knowledge domains, such as to a new document corpora, remains challenging. These raw documents lack verifiable reward signals, making it difficult to apply RL solutions. As a result, the dominant paradigm in this regime is supervised continual learning, which attempt to address catastrophic forgetting by constraining parameters, gradients, or data mixtures during fine-tuning (Ouyang et al., 2022; Guo et al., 2025; Gu et al., 2024). However, these modifications still rely on next-token prediction as their primary supervision signal, treating the preservation of prior behavior as an *auxiliary* objective. Empirically, these approaches fail to reliably mitigate forgetting in continual adaptation.

In this work, we propose **Distillation via Split Contexts (DiSC)**, a context distillation method for continual knowledge adaptation of post-trained LLMs. Rather than treating behavior preservation as an auxiliary regularizer, DiSC uses the KL divergence as its primary objective to achieve behavior preservation. Particularly, DiSC trains the model to match predictions of a teacher model derived from the same model policy, but conditioned on a richer context that contains the new knowledge we want the LLM to learn. Thus, DiSC effectively minimizes behavioral divergence while incorporating new knowledge in the trained student model.

To evaluate DiSC, we run experiments on four post-trained LLMs: QWEN-2.5-7B-INSTRUCT, QWEN-2.5-3B-INSTRUCT , LLAMA3.1-8B-INSTRUCT, and QWEN3-8B. We train for domain adaptation on two domains, KUP (Li &

[1]Department of Computer Science, Cornell University. Correspondence to: Shankar Padmanabhan <sp2583@cornell.edu>.

*Proceedings of the 43rd International Conference on Machine Learning*, Seoul, South Korea. PMLR 306, 2026. Copyright 2026 by the author(s).

[2]Code is publicly shared at https://github.com/shankarp8/distillation-retains-capabilities

Goyal, 2025), a news-style dataset containing synthetic updates to world knowledge, and a biomedical dataset BioASQ (Krithara et al., 2023). We compare DiSC against prior finetuning techniques (e.g., with on-policy data (Chen et al., 2025), token-adaptive loss (Lin et al., 2025), etc.) and knowledge distillation approaches proposed for continual adaptation (Padmanabhan et al., 2023). In addition to domain adaptation, we measure catastrophic forgetting for a suite of 7 standard tasks, including Big-Bench Hard (Suzgun et al., 2022), GPQA (Rein et al., 2023), IFEval (Zhou et al., 2023), MATH-Hard (Hendrycks et al., 2021), MuSR (Sprague et al., 2024), MMLU-Pro (Wang et al., 2024), and the coding benchmark HumanEval (Chen et al., 2021).

**Finding 1: DiSC mitigates catastrophic forgetting during continual knowledge adaptation.** Our results show that DiSC outperforms prior methods on domain adaptation, while incurring negligible degradation on instruction-following, reasoning and other prior capabilities. For instance, when adapting QWEN-2.5-7B-INSTRUCT to a document-level corpus, finetuning degrades instruction following by nearly 15 points and math reasoning by over 25 points. In contrast, DiSC limits degradation on both domains to less than 5 points, while achieving *superior* domain gains. These results establish context distillation-based supervision as an effective and performant technique for continual knowledge updating in post-trained LLMs.

**Finding 2: Existing finetuning based methods fail to alleviate catastrophic forgetting.** In particular, we observe a consistent and robust trade-off between capability preservation and domain adaptation that these methods fail to improve. For instance, on-policy finetuning (Chen et al., 2025), TALR (Lin et al., 2025), and KL regularization result in roughly the same degradation on IFEval, MATH, and HumanEval performance as finetuning, across settings. We found that while finetuning with LoRA often reduces forgetting, it comes at the expense of in-domain performance.

**Finding 3: "Post-training skills" suffer the largest degradation after finetuning for continual adaptation.** To understand task degradation after finetuning, we loosely classify tasks into pre-training and post-training skills depending on whether the improvement between the pre- and post-training checkpoints was minimal or significant respectively. Interestingly, we find that post-training skills, such as instruction-following, math reasoning and coding, show the largest degradation after adaptation with finetuning (e.g. ~8-15 points drop in IFEval across models and finetuning baselines on KUP). On the other hand, pretraining skills that did not see improvement with post-training, such as BBH, GPQA, MMLU and MuSR, report negligible degradation. In fact, the improvement in the task performances between pre- and post-training is strongly correlated ($\approx 0.9$) to its subsequent drop after regular finetuning. These results pro-

vide strong evidence that finetuning reverts the post-trained model back towards its pre-trained behavior.

## 2. Continual Knowledge Adaptation

### 2.1. Problem Definition

We study the problem of continual knowledge adaptation, where the task is to adapt an LLM to internalize new knowledge while mitigating the catastrophic forgetting of existing capabilities. In order to emulate realistic update scenarios, we make two design decisions when formalizing our setting:

**First, we assume that the new knowledge is provided in the form of multi-sentence or multi-paragraph documents**. These documents might reflect updates to world knowledge after the training cut-off date (e.g. news articles), data unseen during prior training or even specialized data for a target domain such as medicine or law. This differs from earlier works in knowledge editing and updating that unrealistically assume that knowledge changes are easily accessible in the form of "factoids" or key-value tuples (Meng et al., 2022; Mitchell et al., 2022a; Meng et al., 2023; Padmanabhan et al., 2023; Dai et al., 2022).

**Second, we focus on adapting post-trained LLMs.** Prior research has traditionally explored adaptation of "raw" documents, as in our setting, for pre-trained models (Yang et al., 2024a; Li & Goyal, 2025; Su et al., 2024). However, post-training significantly improves upon the pretrained LLM on several key dimensions, including instruction-following, reasoning, alignment with human preferences, among others. It is infeasible to adapt the pre-trained model on new knowledge and then repeat this complex and expensive post-training pipeline. Therefore, in this paper, we focus on continual knowledge adaptation of post-trained LLMs, where, in addition to adaptation of new knowledge, both pretraining *and* post-training capabilities must be preserved.

**Notation** Formally, let $M_{\text{post}}$ be a post-trained language model obtained after alignment training of a pretrained model $M_{\text{base}}$. Let $\mathcal{D} = \{d^{(1)}, \ldots, d^{(m)}\}$ be a corpus of documents reflecting the new knowledge. Our task is to design an adaptation procedure that is given $M_{\text{post}}$ and $\mathcal{D}$ as input and produces an updated model $M_{\text{new}}$ that internalizes the new knowledge in $\mathcal{D}$ while preserving the general capabilities of $M_{\text{post}}$. The task setup does not specify any details about how the new knowledge in $\mathcal{D}$ will be tested or what the set of general capability tasks will be during training.

The trained models' performance is measured on two different benchmark suites, reflecting the dual goals of the problem setting: 1) **Knowledge Adaptation**: Let $\mathcal{T}_{\mathcal{D}}$ be tasks that require knowledge from $\mathcal{D}$ to answer correctly, such as question answering. 2) **Capability Preservation**: Let $\mathcal{T}_{\text{gen}}$ be a set of tasks that test for general capabilities

already present in $M_{\text{post}}$, such as instruction following, math reasoning or coding.

We report the trained model $M_{\text{new}}$'s performance on both these task suites, denoted by $\text{Acc}(M_{\text{new}}; \mathcal{T}_{\mathcal{D}})$ and $\text{Acc}(M_{\text{new}}; \mathcal{T}_{\text{gen}})$ respectively. A successful adaptation procedure would yield an $M_{\text{new}}$ that maximizes the adaptation performance $\text{Acc}(M_{\text{new}}; \mathcal{T}_{\mathcal{D}})$ while minimizing the drop in prior capabilities $\text{Acc}(M_{\text{post}}; \mathcal{T}_{\text{gen}}) - \text{Acc}(M_{\text{new}}; \mathcal{T}_{\text{gen}})$.

## 2.2. Standard Finetuning as Likelihood Optimization

Standard fine-tuning with the next-token prediction objective has been commonly used to adapt trained language models to new document corpora (Yang et al., 2024a; Siriwardhana et al., 2024; Tanwar et al., 2025).

$$\mathcal{L}_{\text{FT}} = \mathbb{E}_{d \sim \mathcal{D}} \left[ \sum_{t=1}^{|d|} -\log P(d_t \mid d_{<t}) \right] \qquad (1)$$

This directly encourages the model to fit the target distribution. In order to preserve capabilities of the initial model, most continual learning approaches modify this core objective to include KL (Ouyang et al., 2022; Guo et al., 2025) or weight regularization (Gu et al., 2024), use low learning rates (Lin et al., 2025), update smaller subspaces (Hu et al., 2022), or augment $\mathcal{D}$ with replay data of demonstrations of capabilities that we want to preserve (Rolnick et al., 2019).

However, these additional constraints do not fundamentally alter the learning objective. In fact, we find that they incur similar drops in performance on previously learned post-training capabilities as regular fine-tuning (§5.1). In this paper, we propose **Di**stillation via **S**plit **C**ontexts, a training method that overcomes these shortcomings.

## 3. DiSC: Distillation via Split Contexts

### 3.1. Motivation

We hypothesize that the baseline methods we considered in Section 2.2 fail to preserve model performance due to their use of the next-token prediction loss as the primary task objective. Recent work (Shenfeld et al., 2025) reports that fine-tuning leads to similar catastrophic forgetting when used for capability adaptation, such as verifiable task domains of math reasoning or scientific QA. A key insight from this work was that RL-based adaptation training mitigates forgetting because it favors solutions that minimize the KL distance between the updated and the initial policy.

Note that a similar RL approach is unsuitable for knowledge adaptation of "raw" documents as a well-defined reward signal is lacking. On the other hand, adding KL regularization to standard fine-tuning does not reliably mitigate forgetting during knowledge adaptation across model sizes (Lin et al., 2025). **In DiSC, our key idea is to re-frame the knowl-edge adaptation task as a variant of context distillation, and use KL divergence as the main learning objective.**

First, we explain our intuition using QA as the inference-time task. Consider using model $M$ to generate responses for a target question $q$ under two inference settings. In the first case, we perform *grounded* inference where generation is conditioned on a grounding document $d$ that contains the relevant information about the question $q$. Thus, we sample from $P_M(\cdot \mid d, q)$. In the second case, we remove the grounding document and simply sample from $P_M(\cdot \mid q)$.

Intuitively, if $M$ already contains the knowledge from $d$ needed to answer $q$ in its parameters, we expect that the distributions $P_M(\cdot \mid d, q)$ and $P_M(\cdot \mid q)$ will be close to each other. Conversely, if $M$ does not contain the relevant knowledge, we can encourage it to internalize question-specific knowledge from the document $d$ by minimizing the KL divergence between the following two distributions:

$$M_{\text{new}} = \arg \min_{\tilde{M} \in \mathcal{M}} \text{KL} \left[ P_M(y \mid d, q) \, \big\| \, P_{\tilde{M}}(y \mid q) \right] \quad (2)$$

This intuition forms the basis for DiSC. Note that the above objective is similar to context distillation in earlier works (Snell et al., 2022; Askell et al., 2021). However, we cannot directly use this objective as our continual learning setup does not provide any $(d, q)$ pairs, and instead only provides raw text documents $d$ for training.

In our approach, we do not attempt to generate an exhaustive set of questions that cover all knowledge in this corpus in order to operationalize the above intuition. Instead, we train $M$ to internalize document prefixes, by minimizing the KL divergence between its distributions over a suffix with and without conditioning on the prefix.

Importantly, unlike most works involving distillation (Hinton et al., 2014; Guo et al., 2025), **our "teacher" and "student" models originate from *the same policy*. Therefore, rather than deriving a signal from the distribution of a superior model, we derive our signal from the *same model* with *richer contextual conditioning*.**

### 3.2. DiSC Algorithm

Figure 1 outlines our approach. Let $d = (s_1, s_2, \cdots s_n)$ denote the sentences in a document $d \in \mathcal{D}$. We initialize a model $M_T$ as a frozen copy of the initial post-trained model $M_{\text{post}}$; this will serve as the teacher model and its weights will not be updated during training. Separately, we initialize the trainable student model $M_S$ with the parameters of $M_{\text{post}}$.

Below, we elaborate on the learning objective for each document $d$ in our training loop:

**Step 1: Document Splitting**: For each document $d$ in the

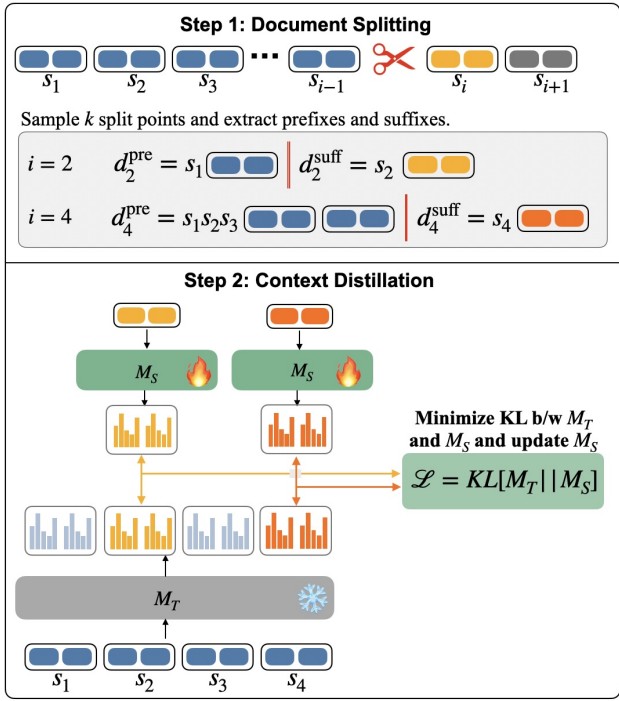

*Figure 1.* Distillation via Split Contexts. First, we sample multiple split points and extract corresponding prefixes and suffixes. Then, we minimize the KL over the suffix distributions with context conditioning (teacher model $M_T$) and without (student model $M_S$). Our training only updates the student model $M_S$.

batch, we get $k$ split points at sentence boundaries; split point $i = 2$ refers to the boundary between sentences $s_1$ and $s_2$. Specifically, we sample $k - 1$ split points uniformly from $\{2, \cdots (n - 1)\}$ and include $n$ as the last split point.

Let $I$ refer to this set of split points. For each split point $i$, we extract a document prefix and a one-sentence suffix from $d$. Specifically, we set the prefix to be all preceding sentences and the suffix to be $s_i$:

$$d_i^{\mathrm{pre}} = s_1 s_2 \cdots s_{i-1} \quad d_i^{\mathrm{suff}} = s_i$$

**Step 2: Context Distillation**: For each suffix $d_i^{\mathrm{suff}}$, we compute the teacher model's distribution conditioned on the corresponding prefix, that is $P_{M_T}(d_i^{\mathrm{suff}} \mid d_i^{\mathrm{pre}})$. We also compute the student model's distribution for the same suffix, that is $P_{M_S}(d_i^{\mathrm{suff}})$, without conditioning on the context. Our document-level learning objective, therefore, is:

$$\mathcal{L}_{\mathrm{DiSC}} = \frac{1}{|\mathcal{I}|} \sum_{i \in \mathcal{I}} \mathrm{KL}\big[P_{M_T}(d_i^{\mathrm{suff}} \mid d_i^{\mathrm{pre}}) \| P_{M_S}(d_i^{\mathrm{suff}})\big] \quad (3)$$

Drawing an analogy with Eq.2 and building on the intuition from §3.1, we can interpret the above equation as encouraging the model to internalize the knowledge in document prefixes by minimizing the KL between the suffix distributions with and without conditioning on the prefix.

We update the student model $M_S$ with this KL-based loss objective. At the end of training, our algorithm returns $M_{\mathrm{new}}$ as the knowledge adapted model.

### 3.3. Efficiency Discussion

Note that Eq. 3 involves computing KL for $|I|$ different suffixes. An inefficient implementation of the objective would require $2 \times |I|$ forward passes for the student and the teacher models combined. However, we emphasize that we can efficiently extract these in a single forward pass of the teacher and student models each.

As shown in Figure 1, we perform one single forward pass with the teacher model using regular attention, and straightforwardly extract the $|I|$ suffix distributions. For the student model, we can construct a new input by concatenating $d_i^{\mathrm{suff}}$ for all $i \in I$ and removing attention connections in between different suffixes. This similarly allows for extracting the $|I|$ suffix distributions in one single forward pass. Therefore, the loss can be computed in two forward passes total.

### 3.4. Connections to prior context distillation literature

Under the broad umbrella of context distillation, prior works (Askell et al., 2021; Choi et al., 2022; Snell et al., 2022; Padmanabhan et al., 2023) differ not only in how contexts and suffixes are obtained but also the underlying goal of the learning algorithm. Table 1 outlines these differences. We characterize differences along three axes: the internalization target (e.g. new knowledge for DiSC), how the context to be internalized is obtained, and the loss tokens for which the KL divergence between the student and the teacher is minimized. As before, we use $d$ to denote a train datapoint.

The first line of work aims to internalize a static context, such as a constitution (Askell et al., 2021) or persona (Choi et al., 2022), for all training points. In contrast, Snell et al. (2022) do not restrict to static contexts and extend context distillation to internalize reasoning processes of the teacher. Here, both the context and the loss tokens are generated by the teacher model. However, neither of these internalize the knowledge in the training datapoint $d$ itself.

The work most closely related to ours is Padmanabhan et al. (2023) which also targets knowledge internalization. However, their method requires generating an explicit *transfer set* of continuations to serve as the suffix, whereas DiSC naturally extracts these from the given document itself. Consequently, DiSC results in a more efficient algorithm by avoiding the expensive generation step during training.

## 4. Experimental Setup

### 4.1. Evaluation Datasets and Benchmarks

As discussed in §2.1, our goal is to internalize new knowledge found in a corpus $\mathcal{D}$ while retaining general capabilities

Table 1. Representative prior works in context distillation.

| Internalization Target | Context | Loss Tokens |
|---|---|---|
| Constitution (Askell et al., 2021) | Static context $c$ for all $d \in \mathcal{D}$ | $d \in \mathcal{D}$ |
| Instructions + Reasoning (Snell et al., 2022) | In-context exs & $c \sim P_{M_T}(\cdot\|d)$ | $y \sim P_{M_T}(\cdot\|d,c)$ |
| New Knowledge (Padmanabhan et al., 2023) | $d \sim \mathcal{D}$ | $y \sim P_{M_T}(\cdot\|d)$ |
| New Knowledge (DiSC) | $d_i^{\mathrm{pre}} \in d \sim \mathcal{D}$ | $d_i^{\mathrm{suff}} \in d \sim \mathcal{D}$ |

of the post-trained model. Below, we describe the datasets $\mathcal{D}$ used in our experiments and their corresponding evaluation suites $\mathcal{T}_{\mathcal{D}}$. We also describe the benchmark tasks $\mathcal{T}_{\mathrm{gen}}$ used to measure general capabilities.

**Evaluating Knowledge Adaptation.** We use 2 document-length datasets $\mathcal{D}$ to continually train our post-trained LMs. These datasets are chosen to capture two orthogonal characteristics of update datasets – new corpora that contradicts existing knowledge and domain-specialized knowledge.

1. **Knowledge Update Playground (KUP)** (Li & Goyal, 2025), which consists of 5,000 synthetic news-style passages that reflect entity-specific changes in world knowledge. We use the multiple-choice QA dataset released with KUP (500 examples) to evaluate our trained models. These questions probe for memorization of new knowledge, particularly if models prioritize recent knowledge over outdated facts and other strong distractors.
2. **Biomedical Dataset BioASQ** (Krithara et al., 2023), a QA dataset with 3,226 documents and 209 associated expert-curated cloze-style questions. In order to evaluate our post-trained models on this dataset, we convert these questions into multiple-choice ones by generating strong distractors using GPT-5. The prompt is in Appendix D.2

**Evaluating General Capabilities.** We measure catastrophic forgetting using a broad set of general capability tasks $\mathcal{T}_{\mathrm{gen}}$. Specifically, we report results on established standard benchmarks Big-Bench Hard (BBH) (Suzgun et al., 2022), GPQA (Rein et al., 2023), IFEval (Zhou et al., 2023), MATH-Hard (Hendrycks et al., 2021), MuSR (Sprague et al., 2024), MMLU-Pro (Wang et al., 2024), and the coding benchmark HumanEval (Chen et al., 2021). These benchmarks probe orthogonal capabilities that are not present in the update dataset, allowing us to isolate catastrophic forgetting from overfitting or domain shift.

## 4.2. Baselines

We compare DiSC against the following baselines:

**1. Fine-tuning (FT) and variants.** Our primary baseline is standard fine-tuning with the next-token prediction objective. We also include baselines that augment this objective:

(i) **+KL regularization** which penalizes deviations from the base model's output distribution during fine-tuning.

(ii) **+LoRA** (Hu et al., 2022), which adapts the model using low-rank updates to a subset of weight matrices while keeping the base model frozen. This constrains parameter updates to a lower dimensional subspace, reducing the extent of model drift during FT.

(iii) **+Rephrase**, which adapts the self-distillation approach introduced in (Yang et al., 2024b) for our setting. It aims to reduce off-policy supervision during finetuning by generating on-policy rephrases of each training document. In particular, we use the initial model $M_{\mathrm{post}}$ itself to generate these document rephrases that are on-policy for $M_{\mathrm{post}}$.

(iv) **Token-adaptive loss reweighting (TALR)** (Lin et al., 2025), which dynamically adjusts the weight of each token's loss to the overall loss based on its predicted probability. Their strategy assigns lower weight to high loss tokens while ensuring that the weight is not concentrated on only a subset of tokens (refer to the original work for details). We follow the weighting curriculum prescribed by (Lin et al., 2025).

**2. Context-Distillation baseline (CD base)** Of the prior works on context distillation, only Padmanabhan et al. (2023) is applicable to our setting. We describe their approach earlier in §3.4. We tune the hyperparameters to our setting. More details in Appendix B.

## 4.3. Models and Training Details

We run experiments on four post-trained models to cover distinct model sizes and families: QWEN-2.5-7B-INSTRUCT, QWEN-2.5-3B-INSTRUCT (Yang et al., 2025b), LLAMA3.1-8B-INSTRUCT (Grattafiori et al., 2024) and QWEN3-8B (Yang et al., 2025a).

All models are trained in FP32 for one epoch. For KL regularization, we set $\beta = 0.1$, a relatively strong regularization value compared to those commonly used in prior work (e.g., Ouyang et al., 2022) in order to mitigate forgetting. For our LoRA variant, we apply LoRA with rank $r = 16$ to the attention and MLP projection layers as per (Schulman & Thinking Machines Lab, 2025). When comparing baseline FT methods in §5.1, we use the same learning rate for all experiments for the same model; we select this as the LR that reports the highest adaptation performance on each respective dataset. We used this strategy as we were unable to run hyperparameter sweeps for each dataset, baseline method, and model combination. We conduct a wider learning rate sweep for standard finetuning, CD base and DiSC in §5.2, to balance both adaptation performance and capability retention. See Appendix B for details.

For evaluation, we use the LM Eval Harness (Gao et al., 2024) to benchmark performance on general capabilities, and custom evaluation scripts for the adaptation tasks. As is standard, we use the respective model's chat template for all evaluations. To reduce variance in domain evaluations, we

*Table 2.* Performance of finetuning baselines and its variants on two adaptation tasks: KUP and BioASQ. We find that standard finetuning generally improves adaptation task performance, but at the cost of substantial degradation of $M_{post}$'s IFEval, Math and Code skills.

| | KUP | | | | | | | | | BioASQ | | | | | | | |
|---|---|---|---|---|---|---|---|---|---|---|---|---|---|---|---|---|---|
| Method | BBH | GPQA | MMLU-P | MuSR | IFEval | Math | Code | KUP | BBH | GPQA | MMLU-P | MuSR | IFEval | Math | Code | BioASQ |
| **QWEN-2.5-7B-INSTRUCT** | | | | | | | | | | | | | | | | |
| $M_{post}$ | 53.69 | 30.29 | 42.89 | 39.81 | 80.70 | 49.85 | 78.66 | 15.46 | 53.69 | 30.29 | 42.89 | 39.81 | 80.70 | 49.85 | 78.66 | 70.19 |
| FT | 52.25 | 28.44 | 38.33 | 40.74 | 65.83 | 24.47 | 69.51 | 40.90 | 54.07 | 30.87 | 37.86 | 43.25 | 75.18 | 35.05 | 73.78 | 77.40 |
| +Rephrase | 51.54 | 28.86 | 37.37 | 43.92 | 63.91 | 28.25 | 71.34 | 42.90 | 52.47 | 29.03 | 37.38 | 39.68 | 71.22 | 32.85 | 71.95 | 76.44 |
| +TALR | 52.70 | 28.86 | 38.90 | 44.18 | 65.83 | 26.51 | 75.61 | 36.90 | 54.05 | 29.28 | 41.30 | 39.15 | 77.70 | 46.98 | 81.1 | 78.37 |
| +KL | 53.39 | 30.03 | 38.28 | 42.99 | 69.78 | 27.57 | 75.00 | 40.10 | 53.81 | 29.95 | 42.50 | 49.40 | 77.94 | 41.27 | 82.32 | 78.37 |
| +LoRA | 52.65 | 29.11 | 39.87 | 42.92 | 75.90 | 43.52 | 78.66 | 35.6 | 48.78 | 30.37 | 39.34 | 40.61 | 74.46 | 38.52 | 75.61 | 76.44 |
| **LLAMA3.1-8B-INSTRUCT** | | | | | | | | | | | | | | | | |
| $M_{post}$ | 50.63 | 29.70 | 37.68 | 38.76 | 85.25 | 20.92 | 70.12 | 20.10 | 50.63 | 29.70 | 37.68 | 38.76 | 85.25 | 20.92 | 70.12 | 79.33 |
| FT | 49.02 | 28.36 | 34.60 | 39.42 | 68.11 | 14.73 | 61.59 | 35.10 | 50.15 | 30.54 | 37.53 | 37.83 | 85.25 | 16.99 | 64.02 | 80.77 |
| +Rephrase | 48.72 | 27.10 | 33.49 | 39.81 | 64.99 | 12.61 | 57.32 | 37.40 | 47.06 | 26.93 | 30.73 | 39.95 | 75.30 | 11.40 | 54.27 | 78.85 |
| +TALR | 50.58 | 27.52 | 35.40 | 38.36 | 73.62 | 14.73 | 62.80 | 32.40 | 48.98 | 29.53 | 31.97 | 41.40 | 81.18 | 13.29 | 64.63 | 82.21 |
| +KL | 50.39 | 27.10 | 33.93 | 38.76 | 64.87 | 16.10 | 60.98 | 33.70 | 49.71 | 27.94 | 32.69 | 37.30 | 80.70 | 13.37 | 60.98 | 81.25 |
| +LoRA | 50.82 | 28.69 | 37.28 | 37.43 | 80.82 | 19.26 | 67.07 | 31.00 | 51.29 | 29.70 | 37.02 | 37.04 | 85.49 | 16.01 | 62.20 | 81.25 |
| **QWEN-2.5-3B-INSTRUCT** | | | | | | | | | | | | | | | | |
| $M_{post}$ | 46.54 | 28.52 | 32.75 | 39.42 | 72.66 | 37.24 | 71.14 | 26.41 | 46.54 | 28.52 | 32.75 | 39.42 | 72.66 | 37.24 | 71.14 | 67.79 |
| FT | 45.74 | 29.53 | 32.79 | 41.80 | 64.39 | 23.87 | 56.10 | 44.57 | 46.66 | 27.77 | 33.63 | 40.74 | 70.86 | 29.68 | 64.63 | 73.08 |
| +Rephrase | 46.43 | 28.52 | 34.15 | 43.92 | 61.03 | 24.55 | 67.07 | 38.06 | 46.00 | 28.36 | 30.74 | 41.40 | 62.71 | 25.08 | 60.98 | 69.71 |
| +TALR | 47.91 | 29.61 | 36.06 | 37.70 | 70.50 | 36.25 | 73.13 | 41.86 | 51.97 | 29.03 | 35.79 | 41.80 | 75.18 | 34.44 | 64.02 | 69.71 |
| +KL | 47.82 | 28.61 | 35.22 | 39.81 | 70.02 | 36.18 | 69.51 | 37.35 | 45.24 | 29.78 | 33.69 | 40.34 | 67.03 | 28.17 | 65.24 | 67.79 |
| +LoRA | 47.86 | 27.94 | 35.85 | 41.40 | 68.47 | 36.33 | 69.51 | 45.00 | 43.46 | 27.94 | 30.30 | 39.15 | 70.86 | 33.76 | 70.12 | 62.50 |
| **QWEN3-8B** | | | | | | | | | | | | | | | | |
| $M_{post}$ | 47.37 | 25.34 | 34.73 | 40.48 | 85.49 | 79.68 | 84.15 | 16.30 | 47.37 | 25.34 | 34.73 | 40.48 | 85.49 | 79.68 | 84.15 | 75.48 |
| FT | 53.88 | 26.43 | 26.89 | 33.99 | 75.54 | 71.90 | 74.39 | 35.40 | 53.46 | 26.01 | 37.48 | 36.98 | 87.89 | 80.82 | 85.37 | 88.69 |
| +Rephrase | 53.46 | 26.01 | 34.53 | 35.45 | 49.88 | 66.09 | 75.00 | 32.20 | 54.57 | 26.01 | 41.58 | 39.81 | 86.93 | 83.01 | 85.37 | 82.21 |
| +TALR | 53.60 | 26.26 | 28.90 | 36.64 | 73.02 | 69.26 | 80.49 | 29.00 | 53.84 | 25.76 | 39.54 | 41.53 | 86.09 | 80.82 | 81.71 | 83.17 |
| +KL | 53.62 | 26.09 | 27.89 | 34.79 | 77.82 | 70.32 | 79.88 | 30.80 | 51.61 | 26.01 | 40.36 | 40.61 | 87.41 | 81.65 | 82.93 | 85.10 |
| +LoRA | 52.63 | 26.01 | 40.67 | 36.38 | 88.01 | 81.27 | 85.98 | 29.90 | 54.76 | 26.17 | 38.31 | 42.86 | 86.45 | 79.58 | 85.98 | 83.17 |

sample 5 answers for each question and use majority vote.

# 5. Results

First, we discuss the performance and limitations of standard fine-tuning and its variants in §5.1. Then, in §5.2, we show that DiSC successfully mitigates forgetting during continual adaptation compared to baselines.

## 5.1. Performance of Fine-Tuning and its Variants

Table 2 reports these results.

**Standard finetuning reliably reports high adaptation performance but at the cost of substantial degradation to general capabilities.** For example, for QWEN-2.5-7B-INSTRUCT, finetuning reports a large gain on KUP adaptation ($15.46 \rightarrow 40.9$), but this is accompanied by sharp drops in instruction following (IFEval:-14.9), mathematical reasoning (Math:-25.4) and coding abilities (Code:-9.15). We find similar trends for LLAMA3.1-8B-INSTRUCT on KUP: FT increases KUP performance by 15 points, but reduces IFEval by -17.1, Math by -6.2 and Code by -8.5. For QWEN3-8B, FT increases KUP performance by 19 points at the cost of reducing IFEval by -10.0 and MATH by -7.80.

While not directly comparable, we observe that finetuning on BioASQ generally leads to lesser catastrophic forgetting than KUP. We attribute this to the different levels of "domain-shift" that these datasets reflect. To elaborate, note that all $M_{post}$ models report very poor performance on KUP and high gains after continual adaptation. On the other hand, BioASQ reports relatively higher initial performance, and lesser improvement after finetuning.

**The highest degradation is generally observed in math, code and instruction-following capabilities.** Notably, MuSR, while also a reasoning task similar to math and code, does not suffer from catastrophic forgetting. We hypothesize that math, code and instruction-following correspond to capabilities that the model acquires during post-training, as opposed to pre-training, and therefore might be more brittle. We verify this hypothesis in §6.1. Interestingly, we observe that finetuning improves QWEN3-8B performance on BBH for both domains by ∼6 points, and MMLU-P for BioASQ by ∼3-7 points. Our analysis in §6.1 shows that these performance numbers after adaptation are similar to that of $M_{post}$'s pretrained checkpoint, providing further evidence that changes to the model after post-training might be brittle under further finetuning.

*Table 3.* Comparing DiSC to standard fine-tuning and CD base following adaptation on KUP and BioASQ. The superscript CP indicates model checkpoints achieving high domain adaptation while experiencing limited forgetting. Across datasets, DiSC consistently reports a stronger trade-off between these two goals.

| Method | KUP | | | | | | | | BioASQ | | | | | | | |
|---|---|---|---|---|---|---|---|---|---|---|---|---|---|---|---|---|
| | BBH | GPQA | MMLU-P | MuSR | IFEval | Math | Code | KUP | BBH | GPQA | MMLU-P | MuSR | IFEval | Math | Code | BioASQ |
| **QWEN-2.5-7B-INSTRUCT** | | | | | | | | | | | | | | | | |
| $M_{post}$ | 53.69 | 30.29 | 42.89 | 39.81 | 80.70 | 49.85 | 78.66 | 15.46 | 53.69 | 30.29 | 42.89 | 39.81 | 80.70 | 49.85 | 78.66 | 70.19 |
| FT (from §5.1) | 52.25 | 28.44 | 38.33 | 40.74 | 65.83 | 24.47 | 69.51 | 40.90 | 54.07 | 30.87 | 37.86 | 43.25 | 75.18 | 35.05 | 73.78 | 77.40 |
| FTCP | 53.58 | 31.80 | 42.01 | 41.14 | 76.74 | 45.24 | 81.71 | 36.3 | 52.66 | 29.78 | 42.67 | 39.02 | 80.46 | 50.53 | 80.49 | 73.08 |
| CD baseCP | 54.42 | 30.62 | 43.02 | 40.08 | 79.38 | 49.24 | 82.32 | 32.10 | 54.89 | 30.45 | 43.41 | 39.95 | 81.65 | 46.83 | 81.10 | 76.92 |
| DiSCCP | 54.07 | 28.27 | 41.27 | 40.48 | 77.70 | 44.34 | 76.22 | 44.18 | 53.81 | 28.86 | 42.10 | 41.27 | 80.82 | 46.75 | 76.83 | 81.73 |
| **LLAMA3.1-8B-INSTRUCT** | | | | | | | | | | | | | | | | |
| $M_{post}$ | 50.63 | 29.70 | 37.68 | 38.76 | 85.25 | 20.92 | 70.12 | 20.10 | 50.63 | 29.70 | 37.68 | 38.76 | 85.25 | 20.92 | 70.12 | 79.33 |
| FT (from §5.1) | 49.02 | 28.36 | 34.60 | 39.42 | 68.11 | 14.73 | 61.59 | 35.10 | 50.15 | 30.54 | 37.53 | 37.83 | 85.25 | 16.99 | 64.02 | 80.77 |
| FTCP | 51.17 | 29.28 | 37.87 | 38.36 | 82.01 | 20.54 | 70.12 | 33.9 | 50.15 | 30.54 | 37.53 | 37.83 | 85.25 | 16.99 | 64.02 | 80.77 |
| CD baseCP | 50.65 | 28.59 | 36.31 | 38.23 | 79.74 | 18.66 | 74.39 | 35.80 | 51.48 | 30.20 | 37.11 | 38.89 | 84.29 | 20.02 | 67.68 | 80.29 |
| DiSCCP | 51.73 | 29.95 | 36.34 | 42.06 | 79.14 | 19.86 | 65.24 | 44.88 | 51.85 | 29.45 | 35.85 | 39.15 | 85.01 | 18.96 | 64.63 | 81.73 |
| **QWEN-2.5-3B-INSTRUCT** | | | | | | | | | | | | | | | | |
| $M_{post}$ | 46.54 | 28.52 | 32.75 | 39.42 | 72.66 | 37.24 | 71.14 | 26.41 | 46.54 | 28.52 | 32.75 | 39.42 | 72.66 | 37.24 | 71.14 | 67.79 |
| FT (from §5.1) | 45.74 | 29.53 | 32.79 | 41.80 | 64.39 | 23.87 | 56.1 | 44.57 | 46.66 | 27.77 | 33.63 | 40.74 | 70.86 | 29.68 | 64.63 | 73.08 |
| FTCP | 48.22 | 30.12 | 35.78 | 39.29 | 69.66 | 32.70 | 75.00 | 38.5 | 46.66 | 28.44 | 32.38 | 39.02 | 71.58 | 37.46 | 65.85 | 64.42 |
| CD BaseCP | 47.82 | 29.11 | 34.78 | 41.27 | 71.58 | 38.07 | 67.68 | 41.5 | 48.10 | 29.45 | 36.32 | 40.74 | 70.74 | 35.95 | 71.34 | 65.38 |
| DiSCCP | 47.80 | 29.61 | 35.40 | 41.40 | 70.02 | 35.95 | 73.08 | 42.27 | 47.80 | 27.18 | 35.99 | 39.81 | 72.42 | 37.08 | 72.56 | 71.15 |
| **QWEN3-8B** | | | | | | | | | | | | | | | | |
| $M_{post}$ | 47.37 | 25.34 | 34.73 | 40.48 | 85.49 | 79.68 | 84.15 | 16.30 | 47.37 | 25.34 | 34.73 | 40.48 | 85.49 | 79.68 | 84.15 | 75.48 |
| FT (from §5.1) | 53.88 | 26.43 | 26.89 | 33.99 | 75.54 | 71.90 | 74.39 | 35.40 | 53.46 | 26.01 | 37.48 | 36.98 | 87.89 | 80.82 | 85.37 | 88.69 |
| FTCP | 56.59 | 25.92 | 34.51 | 37.30 | 83.69 | 81.80 | 80.49 | 32.60 | 53.46 | 26.01 | 37.48 | 36.98 | 87.89 | 80.82 | 85.37 | 88.69 |
| CD baseCP | 54.80 | 26.01 | 39.20 | 41.40 | 75.90 | 80.91 | 83.54 | 28.30 | 54.39 | 26.01 | 40.30 | 41.27 | 85.49 | 81.72 | 82.32 | 85.10 |
| DiSCCP | 55.88 | 26.17 | 39.65 | 39.29 | 83.21 | 77.79 | 82.32 | 37.40 | 53.98 | 26.01 | 42.54 | 41.80 | 83.93 | 80.21 | 84.15 | 89.90 |

**All fine-tuning variants trade off adaptation or capability preservation at the cost of the other without universal improvements.** Both TALR and KL regularization mitigate forgetting in select settings, but these gains are inconsistent across models and datasets. For QWEN-2.5-3B-INSTRUCT trained on KUP, both approaches eliminate forgetting on IFEval, Math and Code compared to finetuning, at the cost of slightly worse domain adaptation. However, on BioASQ, TALR and KL fail to retain their Code and Math capabilities respectively. Added to this, these approaches also fail to mitigate forgetting for QWEN-2.5-7B-INSTRUCT, QWEN3-8B, and LLAMA3.1-8B-INSTRUCT.

Rephrase shows similarly unreliable improvements. For both QWEN-2.5-7B-INSTRUCT and LLAMA3.1-8B-INSTRUCT, FT + Rephrase is the strongest adaptation baseline for KUP (improves over regular finetuning by 2+ points), but causes similar (or higher) drops in Math, IFEval and Code. Moreover, it reports a much lower adaptation result compared to finetuning on BioASQ.

Finally, while LoRA does reduce forgetting compared to standard finetuning (e.g. only -4.80 on IFEval compared to -14.87 w/ finetuning for QWEN-2.5-7B-INSTRUCT), it is most often accompanied by lower domain performance (e.g., 35.6 vs 40.9 for the same model) on KUP.

Overall, these results highlight that none of the variants of finetuning are able to consistently mitigate forgetting while maintaining high knowledge adaptation performance.

### 5.2. Context-distillation methods v/s regular fine-tuning

Next, we study the continual adaptation performance of context-distillation methods, including DiSC. We choose regular FT as the representative method from §5.1.

We report results in Table 3 for the checkpoint that achieves the highest adaptation score while experiencing limited forgetting when varying learning rates. We allow a maximum of ∼5 point drop on each of IFEval, Math and HumanEval. For an apples-to-apples comparison, we use this same selection criterion for all methods. We call these the **capability-preserving** checkpoints, and use the superscript CP to highlight these in the table. In Table 6 (Appendix A), we also report results using the LR that maximizes adaptation performance on KUP for DiSC, as in §5.1.

**DiSC achieves the strongest adaptation gains with minimal forgetting.** Across all models and datasets, DiSC reports the highest adaptation performance among the capability preserving checkpoints, and even outperforms unconstrained finetuning on domain adaptation in six out

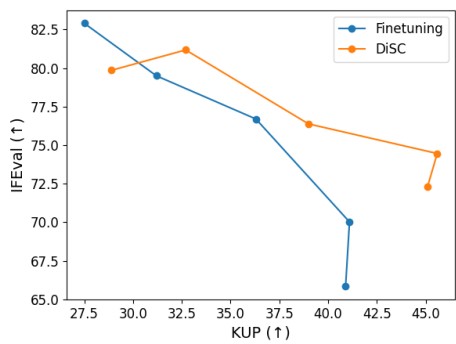

*Figure 2.* IFEval vs KUP performances at different learning rates on QWEN-2.5-7B-INSTRUCT for standard finetuning and DiSC.

*Table 4.* Comparing $M_{\text{base}}$ and $M_{\text{post}}$ models across capabilities. Post-training primarily improves on IFEval, Math and Code.

| Model | BBH | GPQA | MMLU-P | MuSR | IFEval | Math | Code |
|---|---|---|---|---|---|---|---|
| **QWEN-2.5-7B-INSTRUCT** | | | | | | | |
| $M_{\text{base}}$ | 49.47 | 29.11 | 42.08 | 38.10 | 57.19 | 27.42 | 56.10 |
| $M_{\text{post}}$ | 53.69 | 30.29 | 42.89 | 39.81 | 80.70 | 49.85 | 78.66 |
| **LLAMA3.1-8B-INSTRUCT** | | | | | | | |
| $M_{\text{base}}$ | 46.02 | 31.54 | 32.39 | 37.17 | 15.59 | 5.06 | 43.29 |
| $M_{\text{post}}$ | 50.63 | 29.70 | 37.68 | 38.76 | 85.25 | 20.92 | 70.12 |
| **QWEN-2.5-3B-INSTRUCT** | | | | | | | |
| Base | 43.55 | 26.93 | 32.39 | 40.08 | 45.32 | 16.62 | 36.59 |
| $M_{\text{post}}$ | 46.54 | 28.52 | 32.75 | 39.42 | 72.66 | 37.24 | 71.14 |
| **QWEN3-8B** | | | | | | | |
| Base | 56.52 | 28.36 | 47.41 | 38.10 | 61.99 | 26.89 | 55.49 |
| $M_{\text{post}}$ | 47.37 | 25.34 | 34.73 | 40.48 | 85.49 | 79.68 | 84.15 |

of eight settings. For example, on KUP, LLAMA3.1-8B-INSTRUCT, QWEN-2.5-7B-INSTRUCT and QWEN3-8B outperform FT$^{\text{CP}}$ and CD base$^{\text{CP}}$ by a massive 5-12 points, while QWEN-2.5-3B-INSTRUCT outperforms these baselines by 3.8 and 0.8 points respectively. These superior domain adaptation performances are accompanied by on par or better capability preserving performance. For KUP, the 5-12 point improvement in domain adaptation is accompanied by a mean 1.33 point drop in IFEval and Math performances compared to FT$^{\text{CP}}$. For Code, comparing FT$^{\text{CP}}$ and DiSC, we see ~2 point increase for QWEN3-8B (where DiSC$^{\text{CP}}$'s adaptation is also ~5 points better) and ~5 point decrease for LLAMA3.1-8B-INSTRUCT and QWEN-2.5-7B-INSTRUCT (where DiSC$^{\text{CP}}$'s adaptation is $8-12$ points better). Overall, our results provide strong evidence that **DiSC reports a much stronger trade-off between domain adaptation and forgetting compared to baselines.**

We see similarly strong results on BioASQ. Here, DiSC$^{\text{CP}}$ outperforms the adaptation performance of both baselines for all models; an ~5 points improvement for QWEN-2.5-7B-INSTRUCT and QWEN-2.5-3B-INSTRUCT, and between 1-5 points improvement for the other two. These are accompanied by similar trade-offs with capability preserving performances as the KUP dataset.

**Baseline Context Distillation reduces forgetting at the cost of inconsistent domain adaptation.** Baseline Context Distillation (CD base$^{\text{CP}}$) sometimes improves over FT$^{\text{CP}}$ in terms of domain adaptation, beating it in four of eight settings. However, its in-domain performance inconsistently competes with unconstrained finetuning and DiSC. While it matches these methods for LLAMA3.1-8B-INSTRUCT on BioASQ, it dramatically underperforms both for QWEN-2.5-7B-INSTRUCT and QWEN3-8B on KUP. This indicates that, although CD base is a reasonable strategy to mitigate forgetting, it is not well suited to achieve strong domain adaptation gains.

**DiSC is more robust to variance in learning rates compared to regular finetuning.** Table 3 only reports results for the LR that produced the capability preserving checkpoint for each baseline. To further investigate how choice of learning rates impacts trade-offs between adaptation and preservation, we report performance of DiSC and FT trained on KUP for five learning rates (LRs) between 1e-6 and 1e-5 in Fig 2. We observe that as the LR increases, FT exhibits sharp degradation on IFEval, whereas DiSC sees a comparatively more gradual decline. At the highest LR, DiSC incurs roughly half of the forgetting on IFEval as FT (-8.4 vs -14.87) while achieving higher KUP performance. This also indicates that method is less sensitive to small variances in learning rate.

## 6. Analysis

### 6.1. Which Tasks does Finetuning Harm?

Our results in §5.1 showed that IFEval, Math, and Code were the only tasks for which a consistent and substantial degradation in performance was observed after finetuning. We hypothesize that tasks that reflect skills acquired during post-training, i.e. those for which $M_{\text{post}}$ is significantly better compared to $M_{\text{base}}$, are ones that degrade the most during continual finetuning. To evaluate this, we report results for all four base and post-trained models.

Results in Table 4 confirm our hypothesis. The only tasks which see a meaningful improvement after post-training across all four models were indeed IFEval, MATH, and HumanEval. Furthermore, we compute the Pearson correlation between post training gain: $\Delta_{\text{post}}(t) = \text{Instruct}(t) - \text{Base}(t)$ and continual FT change: $\Delta_{\text{FT}}(t) = \text{Instruct}(t) - \text{FT}(t)$ for all tasks $t$. Across all four models we observed a strong correlation between $\Delta_{\text{post}}(t)$ and $\Delta_{\text{FT}}(t)$: 0.83 for QWEN-2.5-7B-INSTRUCT, 0.98 for LLAMA3.1-8B-INSTRUCT, 0.94 for QWEN-2.5-3B-INSTRUCT, and 0.56 for QWEN3-

*Table 5.* Effect of replay on QWEN-2.5-7B-INSTRUCT finetuned on KUP. Including either gold or self-generated replay data degrades capabilities relative to finetuning on KUP alone.

| Train Data | IFEval | MATH |
|---|---|---|
| KUP | 65.83 | 24.47 |
| KUP + GSM8K replay (gold) | 62.59 | 2.95 |
| KUP + GSM8K replay (self-gen) | 57.40 | 7.70 |
| KUP + AlpacaEval replay (gold) | 48.44 | 8.99 |
| KUP + AlpacaEval replay (self-gen) | 49.85 | 20.85 |

8B. Interestingly, this holds *even for tasks for which the base model outperforms the post-trained version*. For instance, QWEN3-8B-BASE outperforms QWEN3-8B at BBH and MMLU-Pro, and in some cases finetuning on unrelated tasks *improves* performance on these benchmarks (e.g., FT on KUP improves BBH $47.37 \rightarrow 53.88$). This suggests that standard finetuning systematically reverts model behavior towards its pretrained checkpoint.

### 6.2. Does Replay Mitigate Forgetting?

Replay is a long-established strategy for mitigating catastrophic forgetting (Rebuffi et al., 2017). We evaluate whether replay can serve as an effective alternative to DiSC for continual knowledge adaptation of post-trained LLMs.

**Setup.** We finetune QWEN-2.5-7B-INSTRUCT on KUP combined with replay data targeting two of the post-training capabilities most affected by adaptation: math reasoning with GSM8K (Cobbe et al., 2021) and instruction following with AlpacaEval (Li et al., 2023). Following Chen et al. (2024), we use a 1:10 replay-to-adaptation ratio. For each target capability, we construct replay data in two ways: (1) *gold*, using ground-truth solution chains, and (2) *self-generated*, by sampling completions from the initial model checkpoint $M_{\text{post}}$, applying rejection sampling based on correctness for GSM8K (Huang et al., 2024).

**Results.** Table 5 reports our findings. Surprisingly, replay *degrades* capabilities substantially compared to finetuning on the adaptation corpus alone. The effect is most pronounced for capabilities related to the replay distribution: including GSM8K replay data reduces MATH performance from 24.47 to 2.95 (gold) and 7.70 (self-generated), while leaving IFEval relatively less affected. Including AlpacaEval replay similarly damages IFEval ($65.83 \rightarrow 48.44$ with gold replay) and also reduces MATH performance, though less severely than GSM8K replay.

**Discussion.** While these results contradict conventional wisdom that replay preserves capabilities, they are consistent with recent findings (Chen et al., 2024; Kotha et al., 2024) that finetuning can induce large catastrophic forgetting on tasks similar to the training distribution. We hypoth-

esize that introducing a secondary training distribution (e.g., GSM8K) over-specializes the model, harming the broader target capability rather than preserving it. We discuss these results in greater depth in Appendix H.

## 7. Related Work

**Model Editing** work aims to update factual or behavioral knowledge in pretrained language models while minimizing unintended side effects. Early approaches such as ROME (Meng et al., 2022) and MEMIT (Meng et al., 2023) perform targeted weight edits to implant new facts but often induce degradation on unrelated capabilities (Gu et al., 2024). Other approaches explore learning update rules directly (e.g., MEND (Mitchell et al., 2022a)) or avoiding weight changes altogether via external memories (Mitchell et al., 2022b). In contrast to direct weight surgery, Padmanabhan et al. (2023) demonstrate that context-based distillation can effectively propagate injected knowledge. In contrast to these works, which study the injection of simple statements, our work focuses on a more realistic continual learning scenario where we update on LMs on *documents* similar to a new corpora.

**Catastrophic Forgetting** has been extensively studied in continual learning for neural networks. Foundational methods include regularization-based approaches such as Elastic Weight Consolidation (Kirkpatrick et al., 2017) and Synaptic Intelligence (Zenke et al., 2017), which constrain updates to parameters deemed important for previous tasks, and distillation-based methods like Learning without Forgetting (Li & Hoiem, 2017). Another widely used strategy is replay, where a subset of prior data is mixed into training to stabilize performance across tasks (Rebuffi et al., 2017; Lopez-Paz & Ranzato, 2017). More recent work highlights the importance of optimization choices and update structure, showing that adaptive optimizers can exacerbate forgetting (Hsu et al., 2019), while sparsity or selective updates can significantly improve retention (Gu et al., 2024). In the context of LLMs, recent approaches such as TALR (Lin et al., 2025) and on-policy finetuning (Chen et al., 2025) aim to balance domain adaptation with general capability preservation. Our results highlight that distillation-based updates offer a strong and underexplored avenue in continual learning.

## 8. Conclusion

We study continual knowledge adaptation for post-trained LMs, which requires learning new knowledge while preserving prior capabilities. Through comprehensive experiments, we show fine-tuning and its variants induce significant forgetting. To address this, we develop a new context distillation-based method and demonstrate substantially stronger adaptation-retention trade-offs than finetuning baselines on four models and two adaptation domains.

## Acknowledgement

We thank Eunsol Choi and the Cornell NLP group for helpful discussions and feedback. This project was partially supported by NSF grant IIS-2433072, and a gift from Google. We gratefully acknowledge use of the research computing resources of the Empire AI Consortium, Inc, with support from Empire State Development of the State of New York, the Simons Foundation, and the Secunda Family Foundation. Shankar Padmanabhan is gratefully supported by a NSF GFRP.

## Impact Statement

This paper presents work whose goal is to advance the field of Machine Learning. There are many potential societal consequences of our work, none which we feel must be specifically highlighted here.

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

*Table 6.* Comparing maximum in-domain performance of DiSC with that of standard fine-tuning

| | QWEN2.5-7B-INSTRUCT | | | | | | | | LLAMA3.1-8B-INSTRUCT | | | | | | | |
|---|---|---|---|---|---|---|---|---|---|---|---|---|---|---|---|---|
| Method | BBH | GPQA | MMLU-P | MuSR | IFEval | Math | Code | KUP | BBH | GPQA | MMLU-P | MuSR | IFEval | Math | Code | KUP |
| $M_{post}$ | 53.69 | 30.29 | 42.89 | 39.81 | 80.70 | 49.85 | 78.66 | 15.46 | 50.63 | 29.70 | 37.68 | 38.76 | 85.25 | 20.92 | 70.12 | 20.10 |
| FT (from §5.1) | 52.25 | 28.44 | 38.33 | 40.74 | 65.83 | 24.47 | 69.51 | 40.9 | 49.02 | 28.36 | 34.60 | 39.42 | 68.11 | 14.73 | 61.59 | 35.10 |
| DiSC$^{\text{Max KUP}}$ | 53.17 | 28.36 | 41.45 | 39.68 | 74.46 | 44.03 | 77.44 | 47.59 | 50.96 | 29.61 | 35.46 | 40.74 | 75.06 | 17.90 | 64.63 | 49.50 |

# A. DiSC vs FT: Maximum Domain Performance

We report the highest recorded performance for DiSC and FT in Table 6. DiSC is able to achieve significantly more domain adaptation while incurring substantially less forgetting on instruction following, math, and code.

# B. Further Training Details

Following existing work on knowledge editing (Meng et al., 2022)(Padmanabhan et al., 2023) as well as recent recommendations for LLM finetuning (Marek et al., 2025), we train with batch size 1 for all experiments. We experimented with larger batch sizes and the results did not change substantially relative to reported trends.

For the main finetuning experiments, we separately tuned learning rate according to what optimized performance on each domain. On KUP, for QWEN-2.5-7B-INSTRUCT and QWEN-2.5-3B-INSTRUCT this resulted in learning rates of 1e-5, for LLAMA3.1-8B-INSTRUCT this resulted in a learning rate of 4e-6, and for QWEN3-8B this resulted in a learning rate of 1.5e-5. For BioASQ, this resulted in a learning rate of 1e-5 for QWEN-2.5-3B-INSTRUCT, 5e-6 for QWEN-2.5-7B-INSTRUCT and QWEN3-8B, and 1e-6 for LLAMA3.1-8B-INSTRUCT.

For the comparisons between DiSC and FT, we conducted a hyperparameter sweep by evaluating both methods on a set of 10 learning rates between 5e-7 and 2e-5: (5e-7, 8e-7, 1e-6, 2e-6, 3e-6, 4e-6, 5e-6, 8e-6, 1e-5, 2e-5). We use softmax temperature $T = 2.0$ for distillation experiments. We experimented with the number of split points $|I|$ to use in DiSC and found $|I| = 5$ achieved good performance empirically, with $|I| > 5$ leading to only marginal improvements at the cost of more compute.

We use the AdamW (Loshchilov & Hutter, 2019) optimizer for all experiments, with weight decay 0.01, betas of $(0.9, 0.999)$ and epsilon of $1e - 8$.

For LoRA experiments, we use a rank of $r = 16$ with a lora alpha of 32, dropout of 0.1, and no bias. We apply this to all linear layers in the transformer blocks.

# C. Evaluation Details

We use the lm-evaluation-harness (Gao et al., 2024) for general capability evaluations. We evaluated every method on each task using greedy decoding and with their native chat template, with few shot examples cast into a multiturn format. For Qwen3, we evaluate all tasks with thinking mode enabled with the exception of BBH and MMLU-Pro. We find that these tasks suffer significant performance decreases with thinking mode enabled (e.g, MMLU-Pro drops from 34.73 to 11.32). Models were cast to bf16 for evaluation.

For KUP and BioASQ evaluations, we sampled five answers from the model per question at temperature 1.0 and used majority vote to decide. We request structured outputs with `<think>` and `<answer>` tags to simplify parsing; we parse answers even when formatting is violated.

# D. Prompts

### D.1. KUP

We include the prompt used for KUP evaluation below.

**KUP Evaluation Prompt**

```
"""You will answer a multiple-choice question about {entity}.
- Put your reasoning in: <think> ... </think>
- Put ONLY the final answer letter (A/B/C/D) in: <answer> ... </answer>
- Do not include any other text outside these tags.

{question}
"""
```

**BioASQ Distractor Generator Prompt**

```
'''You are a careful biomedical distractor writer.
You will be given: (1) a question, (2) the correct answer, and (3) supporting
context.
Return exactly THREE semantically plausible but wrong answer choices. The wrong
answer choices should be sufficiently close to the correct
answer that a knowledgeable test taker might confuse them.
Rules:
- Make them concise noun phrases (a few words).
- Same answer type/category as the correct answer.
- Must be clearly contradicted by or NOT supported by the context.
- Avoid negations like "not", "none of the above", or humorous/absurd options.
- Do not repeat the correct answer with trivial rephrasing.
- Keep them distinct from each other.
STRICT OUTPUT (JSON only):
{
   "distractors": ["<D1>", "<D2>", "<D3>"]'''
}
```

## D.2. BioASQ

We include the prompt used to generate distractor options for BioASQ below. We also include a few samples of options generated by GPT-5 (alongside one correct answer) below.

Finally, we also include the evaluation prompt for BioASQ:

## E. Does KL Predict Forgetting?

[Shenfeld et al. (2025)](#) found that for a simple neural network finetuned on an MNIST-esque task, the average per-token KL loss between the finetuned and initial model on the training data was the best predictor of forgetting on downstream tasks. In order to validate whether this observation extends to our more realistic setting, we measure the per-token KL loss between the initialization model and the finetuned model with each of the baselines we used. We then compute the correlation between KL loss and forgetting on IFEval and MATH for QWEN-2.5-7B-INSTRUCT and LLAMA3.1-8B-INSTRUCT. We report results in Table 7.

For both models on KUP, the average per-token KL loss on the training data exhibits a moderate positive correlation with forgetting on IFEval and MATH. In other words, a larger KL divergence came with more forgetting. How-

*Table 7.* Per-dataset correlation between KL divergence (measured on the train domain) and forgetting magnitude on held-out tasks.

| Model | Domain | IFEval Pearson | MATH Pearson |
|---|---|---|---|
| **LLAMA3.1-8B-INSTRUCT** | KUP | 0.22 | +0.54 |
| | BioASQ | −0.54 | +0.39 |
| **QWEN-2.5-7B-INSTRUCT** | KUP | +0.51 | +0.50 |
| | BioASQ | −0.55 | −0.40 |

ever, for BioASQ, KL divergence often had a *negative* correlation with forgetting on IFEval (with the exception of LLAMA3.1-8B-INSTRUCT on MATH). This challenges the findings from [Shenfeld et al. (2025)](#) that average per-token KL loss is a strong predictor of forgetting. However, we note that our analysis is limited by small sample sizes ($n = 6$ per model/task combination). Future work may be able to investigate this on a larger set.

## F. Memory and Compute Analysis

In this section, we analyze the memory and compute requirements of DiSC and compare them to standard finetuning (FT), finetuning with KL regularization (FT + KL), and the baseline context distillation method (CD base).

---

**BioASQ Questions and Answers Alongside GPT-5 generated options**

```
Question:
How many selenoproteins are encoded in the human genome?

Options:
(A) 23
(B) 24
(C) 26
(D) 25 (correct)

Question:
Which is the receptor for substrates of chaperone-mediated autophagy?

Options:
(A) HSPA8 / HSC70
(B) LAMP2A (correct)
(C) LAMP2B
(D) LAMP1

Question:
Which enzyme is involved in the maintenance of DNA (cytosine-5-)-methylation?

Options:
(A) Dnmt2
(B) Dnmt1 (correct)
(C) Dnmt3a
(D) Dnmt3b

Question:
Which pituitary adenoma is a common cause of infertility in women?

Options:
(A) Corticotroph adenoma
(B) Somatotroph adenoma
(C) Gonadotroph adenoma
(D) Prolactinoma (correct)
```

---

### F.1. Compute

Standard FT requires one forward pass and one backward pass through the document per training step. DiSC, CD base, and FT + KL all require maintaining a frozen teacher copy of the model in addition to the trainable student, and therefore require two forward passes (one for the teacher, one for the student) and one backward pass per training step. The forward-pass overhead of DiSC over standard FT is the cost of preserving capabilities.

Critically, as described in Section 3.3, DiSC achieves its objective in only two total forward passes despite involving $|\mathcal{I}|$ different suffix distributions. This is enabled by (1) extracting the $|\mathcal{I}|$ suffix distributions from a single teacher forward pass over the document, and (2) constructing a single student input by concatenating the $|\mathcal{I}|$ suffixes with attention masked between them, so that all $|\mathcal{I}|$ student distributions

are obtained in one forward pass. A naive implementation would require $2|\mathcal{I}|$ forward passes; our implementation incurs no asymptotic overhead in the number of split points.

### F.2. Memory

All three KL-based methods (DiSC, CD base, and FT + KL) maintain a frozen teacher and a trainable student, and therefore require storing (1) parameters and activations for the teacher, (2) parameters and activations for the student, and (3) gradients and optimizer states for the student. The differences between methods arise in the activation memory of both models, which scales with input length.

**Teacher activation memory.** In DiSC and FT + KL, the teacher executes one forward pass over the document with no need for KV cache retention. In CD base, the teacher

**BioASQ Evaluation Prompt**

```
"Answer the multiple-choice question by reasoning briefly in
<think>...</think> and then giving ONLY the numeral of the correct option
inside <answer>...</answer>.\n\n"
f"Question:\n{question}\n\n"
f"Options:\n{opts_str}\n\n"
"Rules:\n"
" – Put your final choice as a numeral only (e.g., 2) inside <answer>
tags.\n"
" – Do NOT repeat the option text inside <answer>.\n"
" – Choose exactly one option.\n"
```

must execute a forward pass over both the document and a generated continuation, retaining the KV cache to support the continuation generation. CD base therefore incurs higher teacher-side activation memory than DiSC or FT + KL.

**Student activation memory.** DiSC operates on concatenated suffixes ($|\mathcal{I}|$ sentences total, with $|\mathcal{I}| = 5$ in our experiments), CD base operates on generated continuations, and FT + KL operates on the entire document. In practice, KUP documents average 683 tokens ($\sim$22 sentences) and CD base continuations average 603 tokens. DiSC therefore requires student activation memory proportional to approximately 155 tokens ($683 \times 5/22$), versus 603 tokens for CD base and 683 tokens for FT + KL.

Taken together, **DiSC has the lowest memory footprint among the three KL-based methods**. Standard FT, which maintains no teacher, has lower memory requirements than all three, but at substantial cost to capability preservation as documented in Section 5.1.

## G. Ablation on Split Design Choices

DiSC requires two design choices regarding the split: the number of split points $|\mathcal{I}|$, and whether to split at sentence boundaries or at arbitrary positions. We ablate both here, training QWEN-2.5-7B-INSTRUCT on KUP under each variant.

### G.1. Number of Split Points

We vary $|\mathcal{I}|$ from 2 to 7. Table 8 reports domain adaptation performance on KUP for each setting. All values of $|\mathcal{I}|$ outperform the capability-preserving FT checkpoint (FT$^{CP}$ achieves 36.30 on KUP for this model), indicating that DiSC's benefits are robust to this hyperparameter. Within the swept range, performance increases steadily up to $|\mathcal{I}| = 5$ and then declines, which motivated our choice of $|\mathcal{I}| = 5$ in the main experiments.

Intuitively, DiSC *internalizes* the knowledge in a document prefix by matching the teacher's distribution over a corre-

*Table 8.* Effect of number of split points $|\mathcal{I}|$ on KUP performance for QWEN-2.5-7B-INSTRUCT. Performance is robust across the swept range, peaking at $|\mathcal{I}| = 5$.

| Number of split points ($|\mathcal{I}|$) | KUP Accuracy |
|---|---|
| 2 | 38.65 |
| 3 | 39.60 |
| 4 | 40.46 |
| 5 | **44.28** |
| 6 | 43.47 |
| 7 | 42.47 |

*Table 9.* Sentence-boundary splits substantially outperform random-token splits on KUP for QWEN-2.5-7B-INSTRUCT.

| Method | KUP Accuracy |
|---|---|
| DiSC (sentence boundaries) | **44.18** |
| DiSC (random token splits) | 29.90 |

sponding suffix. To fully internalize the knowledge in a document, every part of the document must appear within a prefix for at least one split point. A small number of split points (e.g., $|\mathcal{I}| = 2$) leaves much of the document outside any prefix, while too many split points ($|\mathcal{I}| > 5$) yields diminishing returns and may introduce noisy supervision from very short prefixes.

### G.2. Sentence-Boundary vs. Random Splits

We next ablate the choice of splitting at sentence boundaries. We compare against a variant in which $|\mathcal{I}|$ split points are sampled at random token positions within the document: each prefix consists of all tokens up to the chosen position, and the corresponding suffix consists of all tokens between that position and the next split point. We use $|\mathcal{I}| = 5$ to match the main experiments. Table 9 reports the results.

Sentence-boundary splits substantially outperform random token splits, with a 14-point gap on KUP. We hypothesize that this stems from a distributional mismatch in the student-side input under random splits. Language models are trained predominantly on semantically coherent inputs;

sentences and documents with well-formed beginnings. Under sentence-boundary splits, both the teacher's input (a prefix ending at a sentence boundary, plus a suffix starting at a sentence boundary) and the student's input (a suffix starting at a sentence boundary) are coherent. Under random token splits, the suffix often begins mid-sentence, yielding an incoherent input that the student has no natural pretraining analog for. Minimizing KL between a teacher conditioned on a coherent context and a student conditioned on an incoherent fragment is unlikely to produce a useful gradient signal, and may actively harm training.

These results suggest that respecting linguistic structure in the splitting strategy is important, and that the simplicity of sentence-boundary splits is not just convenient but principled.

## H. Validation for Replay

Because our results run counter to conventional wisdom on replay, we conducted several checks to rule out artifacts. First, we reproduced the results using an independently reimplemented training pipeline and observed consistent behavior. Second, we verified that optimization proceeds as expected: both the KUP and replay losses decrease substantially during training (e.g., $2.08 \rightarrow 1.16$ and $0.98 \rightarrow 0.40$ respectively), and the GSM8K-replay-augmented model achieves $\sim 80\%$ accuracy on its replay training examples (versus 13% for KUP-only training), indicating strong fitting to the replay data rather than failure to learn it. Third, the degradation persists across replay sources (gold vs. self-generated) and across train/eval formatting choices (e.g., with and without chat templates). We additionally inspected model outputs manually and confirmed that the degradation reflects genuinely worse reasoning, not format mismatches.

Importantly, there are several differences between our setting and prior settings where replay has been shown to succeed. Much of the prior literature studies *continual task learning*, where the target capabilities are explicitly defined and representative data from prior tasks is either available or can be approximated. For example, Huang et al. (2024) use past task data to approximate the distribution of prior tasks via demonstrations, implicitly assuming access to representative task questions. In contrast, our setting aims to preserve the *broad* capabilities of a post-trained model, and we cannot assume access to data representative of the post-training distribution that induced those capabilities. As a result, replay is fundamentally limited in our setting: any concrete replay dataset we choose imposes its own distributional pull, which can shift the model away from the post-trained policy along axes other than the one we hoped to preserve.

Replay has also been successfully used to finetune pre-trained checkpoints (Yang et al., 2024a), whereas we finetune post-trained models. The narrower target distribution of a post-trained model may make it more sensitive to distributional drift from replay sources. It is possible that a formulation of replay that better approximates the post-training distribution would yield different results, but we leave this for future work.

