# OpenReview forum: "Updating Parametric Knowledge with Context Distillation Retains Post-Training Capabilities"
_ICML.cc/2026/Conference — ICML 2026 spotlight_

### Official Review · Reviewer_omZD · 2026-03-06

**Soundness:** 3
**Presentation:** 3
**Significance:** 3
**Originality:** 3
**Overall Recommendation:** 4
**Confidence:** 4

**Summary:**

The paper studies how to do continual knowledge updating without catastrophic forgetting of prior capabilities and knowledge. The proposed method DiSC turns documents into samples for KL-based context-distillation. The teacher model is the same as the student, but prompted with a prefix of the raw document. Experiments are on KUP and BioASQ using 3 different standard base models (Qwen-2.5 3B, 7B and Llama-3.1 8B). They compare FT, FT+KL, LoRA, rephrasing, TALR, and a prior context-distillation baseline. FT is shown to improve performance on targeted domains while reducing performance on other domains. DISC is shown to be a strong point on the tradeoff between preserving prior capabilities and improving on new capabilities, sometimes beating FT on new capabilities (e.g. KUP) while consistently beating FT on old capabilities (e.g. IFEval).

**Compliance With Llm Reviewing Policy:**

Affirmed.

**Final Justification:**

The rebuttal addressed a few of my main concerns, particularly the split design and computational trade offs, and improved my confidence in the method's design choices. However, it didn't substantially change my overall evaluation, because some limitations (limited analysis of KL behavior and scope of evaluation) remain. My final recommendation stays therefore at weak accept, as the paper presents a solid and practically relevant contribution with some limitations in depth.

**Key Questions For Authors:**

1. How important are the specific split design choices? How would it work with fewer splits (e.g. middle-only) and non-sentence-boundary splits?
2. On page 8 it seems that the correlation of KL loss with forgetting was unconclusive. Could you run additional experiments or analyses to better understand when KL is predictive and when it is not?
3.  Can you explain the efficiency difference between FT and DiSC? While there is some comparison of learning rates, it would be helpful to see the comparison of much compute goes into the two methods.

**Limitations:**

yes

**Strengths And Weaknesses:**

## Soundness
Strengths:
- The problem is well-motivated and the solution is simple and makes sense.
- Many benchmarks are evaluated to assess performance degradation, and results are fairly comprehensive and consistent. Figure 2 in particular showing the Pareto frontier of IFEval and KUP was important to include.

Weaknesses:
- The choice of splitting at a few random sentence boundaries is not clearly motivated. It would help to have some ablations on number of split points and on sentence-boundary or not in order to determine which parts of the method are most important. If arbitrary-position splits work, this would show some robustness and would be nice to see.

## Presentation
Strengths:
- The paper is well written. The Results and Analysis sections were structured well.

Weaknesses:
- I would recommend removing emojis from Figure 1. It is also not completely clear from the diagram alone what is going on.

## Significance
Strengths:
- The problem of continual updates without catastrophic forgetting is quite important.
- Context distillation works off of established training approaches and libraries without too much effort, making it potentially a part of frontier model training recipes.

Weaknesses:
- It is unclear how important factual knowledge updates are compared to other language model capabilities. The paper focuses only on knowledge QA.

## Originality
Strengths:
- While there have been several recent works exploring context distillation and on-policy self-distillation as a means for retaining capabilities during continued training, these have all been released concurrently to the ICML submission deadline.

Weaknesses:
- Context distillation and all other components have existed prior to this paper and the novelty is more in the application and analysis.

---

> ### Author Rebuttal · Authors · 2026-03-31
>
> Thank you for your positive assessment of our work. We address all key questions, and main concerns from weaknesses below.
>
> **Q1: How important are split design choices?**
>
> First, we include an ablation on the number of split points, testing 2 through 7 sentences with Qwen2.5-7B-IT trained on KUP. We show the results below. We find that domain adaptation accuracy is quite high, outperforming the capability preserving checkpoint of standard FT for all choices of number of split points. Comparing these, we observe that accuracy steadily increases with the number of sentences up to 5 sentences (our choice in experiments in the paper), and then decreases.
>
> | Sentences | Accuracy |
> |-----------|----------|
> | 2 | 0.3865 |
> | 3 | 0.396 |
> | 4 | 0.4046 |
> | 5 | **0.4428** |
> | 6 | 0.4347 |
> | 7 | 0.4247 |
>
> Intuitively, we understand our technique as “internalizing” knowledge from the prefix by matching the KL of the suffix sentence. To internalize knowledge from the entire document, all parts of the document must be part of a prefix at some point. Therefore, having only a middle split point would not work well using our approach.
>
> Next, we provide results for DiSC when implemented with random split points rather than at sentence-boundaries, as suggested. Specifically, we select |I| random token positions, with each prefix being everything up to that token in the document and the suffix being the tokens between that split point and the next. We use |I|=5, as in sentence-level DiSC. We show the results below.
>
> | Method | KUP |
> |----------------------|-------|
> | DiSC (sentence boundaries) | 44.18 |
> | DiSC (random splits) | 29.9 |
>
> Empirically, we find that DiSC using random split points reports substantially poorer domain adaptation performance than DiSC with sentence-level boundaries. This could be due to multiple reasons. Language models are largely trained on semantically coherent data, e.g. sentences with a coherent beginning. When we use random splits, the prefix is coherent (as before) but the suffix often starts with a left-truncated part of a previous sentence, resulting in an incoherent input. We hypothesize that reducing KL between a model that sees a well-formed coherent input (teacher) and incoherent input (student) may not be ideal.
>
> **Q2: KL as predictor of forgetting**
>
> We only have checkpoints corresponding to our baselines (n=6). We do not believe these are enough data points to make conclusive statements. However, we included this discussion of KL as a predictor of forgetting to provide a counterpoint to prior work that established this for MNIST [1].
>
> We find the Reviewer cNmu's hypothesis that the degree to which KL is a reliable predictor could be a function of the KL divergence observed after training to be very interesting. We refer the reviewer to our response to Reviewer cNmu for more discussion on this. However, as we point out in that response, we are unable to make reliable conclusions due to the size of our sample.
>
> **Q3: Efficiency Comparison between FT and DiSC**
>
> Compute: FT requires 1 forward and 1 backward pass per training run, as normal. DiSC, on the other hand, requires 2 forward passes (1 for the teacher, 1 for the student) and 1 backward pass per training run. Therefore, DiSC incurs more forward pass compute than FT.
>
> Memory: DiSC requires maintaining two copies of the model: the frozen teacher and the trainable student. As such, it requires storing the parameters and activations for both models, as well as the gradients and optimizer states for the student model.
>
> On the other hand, FT (without KL regularization) only maintains a single model (no teacher) thereby requiring only one set of model parameters, activations, gradients, and optimizer states. Therefore, FT requires less memory than DiSC. However, this comes at the cost of capability preservation.
>
> We also compare DiSC’s memory requirements compared to our baseline context distillation method (CD base), and FT w/ KL regularization in our response to Reviewer cNmu, showing that DiSC incurs less memory than those alternatives.
>
> **Weakness: Main Focus is Knowledge QA**
>
> Knowledge editing and updating is a well-studied and well-established task in literature [2,3,4]. Due to how challenging it is, prior work has mostly studied this for factoids, but we study it in a more realistic setting where new knowledge is provided as document corpora. We emphasize that our goal is to update parametric knowledge, and not improve a skill that is amenable to RL-based training (e.g., math), as the latter has been studied extensively in prior works [1].
>
>
> 1. Shenfeld, S. et al. RL’s Razor: Why Online Reinforcement Learning Forgets Less. arxiv 2025.
> 2. Mitchell, E et al. Fast model editing at scale. ICLR 2022.
> 3. De Cao et al. Editing factual knowledge in language models. EMNLP 2021.
> 4. Meng, K et al. Locating and editing factual associations in GPT. NeurIPS 2022.

---

> > ### Author Rebuttal · Reviewer_omZD · 2026-04-02
> >
> > My concerns are partially resolved. The additional ablations on the split and the discussion of tradeoffs with FT were helpful.
> >
> > However, I agree that the discussion around KL as a predictor of forgetting is not entirely conclusive, and I still view the evaluation scope of knowledge QA as a bit narrow, so practical significance can't fully be established.

---

> > > ### Author Response · Authors · 2026-04-07
> > >
> > > Thank you for the thoughtful follow-up and for acknowledging that the additional ablations and efficiency discussions were helpful.
> > >
> > > **"However, I agree that the discussion around KL as a predictor of forgetting is not entirely conclusive”**: We agree with this statement, and explicitly and intentionally present it as such in both the paper and rebuttal. Therefore, we are confused at this being viewed as the main weakness as this is far from being a core contribution of our work (we discuss core contributions in the next paragraph). To contextualize this particular KL experiment – a recent paper [1] showed that KL can serve as a predictor for forgetting and is frequently cited as proof for this claim despite the original paper acknowledging its limited experimental scope and warning against generalizing to other domains and frontier models. The experiment in our paper provides a counterexample to this widely cited claim, which we believe is useful for the research community.
> > > The central contribution of the paper is our DiSC method  which uses ideas from context-distillation to achieve continual learning adaptation, and empirical demonstration of DiSC’s superior trade-offs between adaptation and capability preservation across multiple models and benchmarks.
> > >
> > > The KL analysis, by contrast, is a tertiary observation, and our conclusions are appropriately cautious. We therefore believe that it should not substantially affect the overall assessment of the paper’s technical contribution.
> > >
> > > [1] RL's Razor: Why Online Reinforcement Learning Forgets Less, Shenfeld et al. 2025
> > >
> > > **“I still view the evaluation scope of knowledge QA as a bit narrow, so practical significance can't fully be established.”**: We respectfully disagree that the evaluation scope is narrow in a way that limits practical significance. As the rebuttal process does not allow us to seek clarifications on the reviewer’s exact concern, we interpret this weakness as meaning one of two things:
> > >
> > > (1) Practical significance of knowledge updating is limited without considering other capabilities like reasoning: First, we strongly disagree that updating the factual knowledge of an LLM has no practical significance. Knowledge editing and updating has a rich history in language modeling research because it addresses a key limitation of current LLMs which is that they are static artifacts while world knowledge continues to evolve. Therefore, it is crucial to continuously adapt these models to reflect up-to-date information or specialize to new domains.
> > >
> > > Furthermore, as we mention in the original rebuttal, prior works have (solely) focused on improving reasoning capabilities like math using RL-based approaches. However, these techniques do not transfer to knowledge, which motivates developing methods for the factual domain. As the techniques involved for knowledge update (our work) and reasoning (prior works) are so different, we (and prior papers) view these as complementary problems and addressing both is out of scope for a single paper.
> > >
> > > (2) Using QA-style evaluation for factual knowledge learning is limited: QA is the standard and most direct method for evaluating whether models have internalized factual knowledge from a corpus, and is widely used in prior work on knowledge editing and continual learning (e.g., [1,2,3]).
> > >
> > > In addition, we complement knowledge QA with a broad suite of general capability benchmarks (IFEval, MATH, HumanEval, BBH, GPQA, etc.), ensuring that our evaluation captures both axes of the continual adaptation problem. We believe this combination provides a comprehensive and practically meaningful evaluation of the setting we study.
> > >
> > > [1]Wang, Y., Wang, M., Manzoor, M.A., Liu, F., Georgiev, G.N., Das, R.J., Nakov, P. Factuality of Large Language Models: A Survey EMNLP, 2024.
> > >
> > > [2] Fang, J., Jiang, H., Wang, K., Ma, Y., Shi, J., Wang, X., He, X., Chua, T. AlphaEdit: Null-Space Constrained Knowledge Editing for Language Models. ICLR, 2025.
> > >
> > > [3] Zhong, Z., Wu, Z., Manning, C.D., Potts, C., Chen, D. MQuAKE: Assessing Knowledge Editing in Language Models via Multi-Hop Questions. EMNLP, 2023.

---

### Official Review · Reviewer_cNmu · 2026-03-12

**Soundness:** 3
**Presentation:** 3
**Significance:** 3
**Originality:** 3
**Overall Recommendation:** 5
**Confidence:** 4

**Summary:**

The paper studies continual training of LLMs and proposes a method to retain post-training capabilities during further training. The key idea is to match the model's outputs to its earlier in-context predictions while adapting to new training data through self-distillation. Experimental evaluation suggests that the approach helps in preserving skills like reasoning and instruction-following acquired during post-training.

**Compliance With Llm Reviewing Policy:**

Affirmed.

**Final Justification:**

The rebuttal addressed my main concerns satisfactorily. The memory analysis provided concrete numbers confirming DiSC's efficiency advantage, the KL divergence discussion offered a plausible explanation for the differing correlation signs, and the MuSR behavior is well explained within the method's own framework. Remaining open points are reasonably scoped out as future work.

The weaknesses I identified are relatively minor in the context of the overall contribution, and the rebuttal has strengthened my confidence in the paper. I am therefore raising my score from 4 to 5.

**Key Questions For Authors:**

1. What are the memory implications of DiSC compared to CD base and FT+KL?
2. Can you comment on the correlation between KL divergence loss and forgetting w.r.t. KUP and BioASQ? Do you think that the average KL loss between pre- and post-distilled models could still be a good predictor if we take into account the magnitude of domain shift? Perhaps this influences the correlation signs (positive for KUP, negative for BioASQ).
3. The results suggest that larger models may suffer more from forgetting (although it may be a fluke due to a small variation in model size). Do you expect that this would hold if the experiments were scaled to larger models? If so, do you have a possible explanation for this?

**Limitations:**

There is no systematic discussion of the limitations.

**Strengths And Weaknesses:**

### Strengths
- The paper addresses an important problem of catastrophic forgetting during continual training of LLMs, particularly the loss of post-training capabilities.
- The exposition is generally clear and easy to follow.
- The proposed context distillation mechanism is conceptually simple, and the efficient implementation described in the paper suggests that it is practical to apply the method during training.
- The empirical results are strong and show consistent improvement over baselines and similar approaches.
- The analysis of domain shift effects (large shift in KUP vs. small shift in BioASQ) is interesting, especially the observation that smaller domain shifts appear to lead to less forgetting with standard fine-tuning. I think this could potentially offer a good explanation of why such approaches "override" capabilities learned during post-training (typically reasoning, coding, or instruction following).

---

### Weaknesses
- The part of the evaluation with respect to target datasets (KUP, BioASQ) is somewhat narrow, since both datasets are QA-style memorization benchmarks. Yes, the datasets are large, but I think that including a dataset focused on another type of problem would strengthen the empirical validation of the method.
- The compute implications of the method are not analyzed. Since the paper emphasizes the efficient implementation of DiSC, I think it's important to compare it to standard distillation setups. I presume that the runtime cost should be similar to the standard setup, but the memory overhead of the approach remains unclear.
- Some results would benefit from further interpretation, at least a speculation on the reasons why. For instance, it is not clear why MUSR behaves differently from other tasks, or why larger models appear to experience more forgetting, as the results seem to suggest.
- Additional evaluation dimensions could strengthen the paper. For example, I think that calibration is an important component to evaluate, especially in distillation setups, which can be detrimental to calibration. I miss the comparison of calibration between the pre- and post-distilled models. Additionally, while less critical, it would be interesting to examine how the method affects alignment-related behaviors, such as safe responses or refusal.
- I miss a discussion of the method's limitations. For instance, the approach relies on preserving the model's previous in-context behavior through distillation, which implicitly assumes that the teacher's behavior is itself desirable and stable.

---

#### Comments
- Figure 2: it is unclear which learning rate corresponds to each point. Is the x-axis sorted by learning rate?
- L397--L398: missing punctuation and space after the reference (Mitchell et al., 2022b)
- L422--L426: sentence beginning with "Our evaluation builds directly on this literature by systematically..." appears to be missing a verb
- The MATH dataset reference (Hendrycks) URL leaks out of the page margin

---

> ### Author Rebuttal · Authors · 2026-03-31
>
> Thanks for your positive assessment of our work. We address the key questions first, and then the weaknesses.
>
> **Q1: Memory Implications of DiSC vs CD Base vs FT + KL**
>
> All three (DiSC, CD Base, and FT + KL) require maintaining two copies of the model: the frozen teacher and the trainable student. As such, all three require storing (1) params and activations for the teacher, (2) params and activations for the student, (3) gradients and optimizer states for the student. The main memory differences come from the activations part of (1) and (2).
>
> For the teacher model: DiSC and FT + KL require the teacher model to execute one forward pass over the document (no KV cache). For CD Base, we execute the teacher forward pass over the document and continuation and need to also maintain the KV cache. Therefore, the teacher model in CD Base requires more memory.
>
> For the student model: Student activation memory scales with input length: DiSC uses concatenated suffixes (|I| sentences, with |I|=5 in our implementation), CD Base uses the continuation, and FT + KL uses the whole document.
>
> In practice, each document in KUP has on average 683 tokens (~22 sentences) and CD base continuations average 603 tokens. Therefore, DiSC requires student activation memory proportional to ~155 (683*5/22) tokens, 603 for CD Base, and 683 for FT + KL
>
> Taken together, DiSC has the lowest memory requirements of the three that use KL.
>
> Thanks for raising this point, we will add this discussion to the paper.
>
> **Q2: KL Divergence Correlation**
>
> Your suggestion that KL can be a good predictor if we factor in the domain shift is very interesting! The per-token KL between checkpoints before and after training is much larger for KUP compared to BioASQ (e.g., 1.01 nats for FT on KUP compared to 0.157 nats for FT on BioASQ). So, it is possible that KL is better correlated with forgetting when the training impacts distributions substantially. E.g., for KUP, we see positive correlations between KL and IFEval/Math performances for both models. However, it may be a less reliable indicator when KL divergence is negligible.
>
> Unfortunately, as we acknowledge in the paper, we do not have enough data points to conclusively answer this.
>
> **Q3: Do larger models forget more?**
>
> We do observe this trend within our results, but as you point out, we must be cautious in drawing strong conclusions given limited size variation (3B, 7B, 8B). We will note this as an open question for future work.
>
> **W1: Narrow Evaluation (only QA-style adaptation benchmarks)**
>
> We do not completely understand the reviewer’s point here and would appreciate some clarification. In our problem definition, we emphasize that our goal is to update parametric knowledge, i.e. memorize new knowledge, and not improve a skill that is amenable to RL-based training (e.g. math) as the latter is studied in prior works.
>
> Within knowledge benchmarks, it is standard to use short-form QA or MCQ style benchmarks for reliable evaluation [1,2]. Additionally, we emphasize that our capability preservation evaluation is broad: for instance, IFEval, MATH, and HumanEval require free-form generation.
>
> **W2: Compute (Answered in Q1)**
>
> **W3: MuSR performance does not decrease after continued training, as is the case with other reasoning tasks.**
>
> We explain this through our analysis in Section 6.1. We hypothesized that continued training primarily hurts capabilities improved during post-training by reverting the model back to its pretrained checkpoint behavior. We find this to be consistent for all tasks and models we consider. Therefore, as the performance on MuSR does not increase substantially from the pretrained to post-trained checkpoint for any three of the models (e.g., 38.1 for Qwen2.5-7B vs 39.8 for Qwen2.5-7B-Instruct), it follows that the performance on MuSR is not hurt after continued training.
>
> **W4: Calibration/Alignment Evaluation**
>
> We believe that this is outside the scope of our current paper, but it would be an interesting avenue for future work!
>
> **W5: Approach assumes that initial checkpoint capabilities are “desirable and stable”.**
>
> We use adaptation and preserving capabilities of the initial checkpoint as the dual goals for our continued training (CT). The goal of the CT is NOT to improve on the models' prior capabilities, such as math. It is to improve its learning of new knowledge in a corpus, but without impacting the prior capabilities/behaviors. As such, this is a very standard setup for continued training. Furthermore, eliminating undesirable behaviors from the post-trained model is a separate problem statement, unrelated to learning new factual knowledge, and would likely require a different approach to tackle. Therefore, we believe that it is outside of the scope of this work.
>
> 1. Rajpurkar, P. et al. SQuAD: 100,000+ Questions for Machine Comprehension of Text. EMNLP 2016.
> 2. Liang, P. et al. Holistic Evaluation of Language Models. arXiv 2022.

---

> > ### Author Rebuttal · Reviewer_cNmu · 2026-04-05
> >
> > Thank you for your responses. The memory analysis is clear and convincing to me. The explanation of MuSR behavior is consistent with the method's own analysis and resolves my concern. The remaining weaknesses around calibration and alignment evaluation are reasonably scoped out as future work.
> >
> > My main concerns have been addressed, and I am raising my score to 5.

---

> > > ### Author Response · Authors · 2026-04-07
> > >
> > > We thank the reviewer for their thoughtful re-evaluation and engagement. We’re glad that the clarifications addressed your concerns.

---

### Official Review · Reviewer_8P1T · 2026-03-15

**Soundness:** 3
**Presentation:** 3
**Significance:** 3
**Originality:** 3
**Overall Recommendation:** 4
**Confidence:** 4

**Summary:**

When learning a new task, it is commonly known that the model forgets an old task. The authors propose DiSC, where the model performs next token prediction on a set of tokens to match the logits of a teacher model that is allowed to condition on a preceding context. The core intuition is that if the models do not disagree when conditioning on the context, the model will not update its weights. The results show that this method achieves a better tradeoff between learning the new task and forgetting the old task relative to standard baselines like fine-tuning with KL, LoRA, etc.

**Compliance With Llm Reviewing Policy:**

Affirmed.

**Final Justification:**

Explained in followup

**Key Questions For Authors:**

listed above

**Limitations:**

yes

**Strengths And Weaknesses:**

Strengths
1. The paper considers an important problem of learning a new task without forgetting an old task
2. The paper measures performs across multiple tasks/models with multiple representative baselines
3. The idea is conceptually quite nice and operationalized practically.
4. The writeup of the algorithm is nice and I understood it quickly

Weaknesses
1. From my understanding of catastrophic forgetting, the most competitive baseline is replaying samples from the previous stage of training. In my experience, this often causes very little damage in knowledge learning and resolves most forgetting. Is it possible to introduce this as a baseline? Though I know one might not have data from the previous stage, there are works showing that simply generating data from the current model serves as good replay data (examples: https://arxiv.org/abs/2403.01244, https://aclanthology.org/2024.naacl-long.10.pdf, https://arxiv.org/abs/1705.08690)
2. This paper is pitched as reducing forgetting. But it looks like DiSC is a performance improvement itself as well? Like looking at the tradeoff plot in Figure 2, isn't it moreso getting better peak adaptation performance rather than peak forgetting reduction? This is a rather odd thing for me to call a weakness since its impressive to get this gain, but perhaps the more impressive part of this paper is improving adaptation, not reducing forgetting?

Happy to increase my score if the method outperforms a reasonable replay baseline and I get clarification on point 2 (and if I am right about point 2, perhaps the paper needs to reframe its results to be more accurate)

---

> ### Author Rebuttal · Authors · 2026-03-31
>
> **W1: Missing Replay Baseline**
>
> We thank the reviewer for the suggestion. We agree that replay is often a strong baseline and conduct additional experiments with replay.
>
> We collect replay data using: (1) gold solution chains, (2) sampled from the initial model checkpoint (as suggested, used in [3]) with rejection sampling based on correctness for math. We did this for two post-training capabilities: math reasoning using GSM8K and instruction following using AlpacaEval (AE). We finetune Qwen2.5-7B-IT on KUP combined with this replay data, using 1:10 replay ratio as per [1].
>
> Our results (Table below) are very surprising (they are verified across multiple runs and hyperparameter choices). We observe that replay degrades capabilities drastically compared to FT on the adaptation corpus alone. This is particularly true for related capabilities: including GSM8K data (either gold or self-generated) harms MATH performance significantly, but does not affect IFEval performance as much. Including AE data significantly harms IFEval performance, but it also harms MATH performance (although less than including GSM8K data). We observed even poorer results w/ replay on the LLaMA-3.1-8B-Instruct model.
>
> We manually inspected the outputs to confirm that these results were not due to mismatches in answer format or any such reason, and confirmed that they were actually due to worse reasoning capabilities.
>
> We realize that our result contradicts conventional wisdom that replay preserves capabilities. First, we highlight that our results are consistent with recent findings [1,2] that finetuning can cause large catastrophic forgetting for tasks similar to the training distribution. In our case, we hypothesize that replay introduces a secondary training distribution (e.g., GSM8K) which can over-specialize the model, harming the target capability rather than preserving it.
>
> Second, there are other important differences between our setting and most prior settings that study replay. One line of these study continual task learning, where the objective is to learn a set of tasks in succession. There, the target capabilities to preserve are explicitly defined and representative data from prior tasks is either available or can be approximated. E.g., in the first work you cite [3], past data is used to approximate the underlying distribution of prior tasks via demonstrations, which implicitly assumes access to representative task distribution and train questions. In contrast, our setting aims to preserve broad capabilities of post-trained models and we cannot assume access to anything about the data distribution that induced those capabilities. As a result, replay is fundamentally limited in our setting.
>
> Replay has also been used successfully to finetune a pretrained chkpt, whereas we finetune a post-trained model. This could also potentially explain the difference in replay results.
>
> It is possible that there is a formulation of replay that better preserves capabilities, perhaps by finding a better approximation of post-training distribution, but we leave this for future work.
>
> | Train Data | IFEval | MATH |
> |----------------------------------------|--------|-------|
> | KUP | 65.83 | 24.47 |
> | KUP + GSM8K replay (gold) | 62.59 | 2.95 |
> | KUP + GSM8K replay (self-gen) | 57.4 | 7.7 |
> | KUP + AE replay (gold) | 48.44 | 8.99 |
> | KUP + AE replay (self-gen) | 49.85 | 20.85 |
>
> **W2: Is DiSC improving adaptation more than reducing forgetting?**
>
> We agree that DiSC often improves over FT adaptation performance in addition to reducing forgetting. In particular, Figure 2 shows that DiSC shifts the overall tradeoff frontier to the right, achieving better or comparable domain adaptation while incurring less degradation on past capabilities.
>
> We consciously chose to highlight the capability preservation property of DiSC in order to avoid overstating its capabilities. We frame the paper around reducing forgetting because this improvement is the most consistent across models and settings. In contrast, gains in peak adaptation performance are slightly model-dependent. For e.g., on Qwen2.5-3B-Instruct, DiSC is comparable but does not surpass FT in peak domain performance (adaptation acc of DiSC=42.27, FT=44.57), but does substantially reduce forgetting (MATH performance of DiSC drops by only 1.29 points, compared to 13.37 points for FT).
>
> Overall, we view DiSC not purely as a method for improving capability preservation/retention, but as improving the adaptation-retention tradeoff, often yielding gains on both axes. Based on the reviewer’s suggestion, we will edit the paper to center improvement in domain adaptation more.
>
> 1. Chen, H. et al. Continual Memorization of Factoids in Language Models. arXiv 2024.
> 2. Kotha, S. et al. Understanding Catastrophic Forgetting in Language Models via Implicit Inference. ICLR 2024.
> 3. Huang, J. et al. Mitigating Catastrophic Forgetting in Large Language Models with Self-Synthesized Rehearsal. ACL 2024.

---

> > ### Author Rebuttal · Reviewer_8P1T · 2026-04-03
> >
> > Thanks for addressing my weaknesses/questions! To be honest, I am quite suspicious about the replay findings: in my experience and the vast majority of papers, replay doesn't lead to such a significant performance detriment, so its a bit odd to me it hurts as much as your table suggests. Nonetheless, the procedure you described seems like a reasonable faith attempt, so I will update my score to Weak Accept. Again, I'd suggest looking into the replay baseline + general training recipe since something feels off.

---

> > > ### Author Response · Authors · 2026-04-07
> > >
> > > We thank the reviewer for their re-evaluation and continued engagement.
> > >
> > > We agree that the magnitude of degradation induced by replay is surprising. To ensure this is not an artifact, we conducted several validation checks even beyond the rebuttal deadline, and confirmed that our results hold up.
> > >
> > > First, we reproduced the results with an independently reimplemented training pipeline and observed consistent behavior.
> > >
> > > Second, we verified that optimization is functioning as expected: both KUP and replay losses decrease substantially during training (e.g., 2.08 -> 1.16 and 0.98 -> 0.40), and the GSM8K replay-augmented model achieves ~80% accuracy on replay training samples (vs. 13% for KUP-only), indicating strong fitting to the replay data.
> > >
> > > Third, the degradation persists across a wide range of settings, including replay sources (gold vs self-generated), and train/eval formatting choices (e.g., chat template vs no chat template).
> > >
> > > Taken together, these results indicate that the observed degradation reflects a genuine phenomenon in our setting rather than a training or evaluation artifact.

---

### Official Review · Reviewer_nj21 · 2026-03-15

**Soundness:** 3
**Presentation:** 3
**Significance:** 3
**Originality:** 3
**Overall Recommendation:** 5
**Confidence:** 4

**Summary:**

This paper studies continual knowledge adaptation for post-trained LLMs: learning new domain knowledge from documents while retaining post-training capabilities such as instruction following, math, and coding. The proposed method, DiSC, trains a student model to match a frozen teacher’s predictions on sentence suffixes, where the teacher has access to the document prefix and the student does not. Experiments on KUP and BioASQ across Qwen-2.5-7B, Qwen-2.5-3B, and Llama-3.1-8B show that DiSC gives a better adaptation-retention tradeoff than standard fine-tuning variants and a prior context-distillation baseline.

**Compliance With Llm Reviewing Policy:**

Affirmed.

**Key Questions For Authors:**

1. How does DiSC perform when combined with replay data (e.g., mixing a small fraction of instruction-following or math data during training)? This is a natural and potentially complementary strategy that is not explored.

**Limitations:**

yes

**Strengths And Weaknesses:**

## Strengths:

- The paper presents a simple and well-motivated formulation. Recasting continual knowledge adaptation as context distillation is intuitive, and the teacher-student setup is easy to understand. The method is conceptually clean and avoids unnecessary complexity.

- The implementation is also practically appealing. Although the objective could appear expensive at first glance, the paper shows how to compute it with only two forward passes by batching the relevant suffix computations efficiently. This makes the approach more realistic for large-scale use.

- The main empirical result is strong: DiSC achieves a better adaptation-retention tradeoff than the compared alternatives, especially on the larger models. In particular, it improves domain adaptation substantially while keeping degradation on instruction-following, math, and coding benchmarks much smaller than standard fine-tuning.

## Weaknesses

- The paper does not include a replay baseline, which is a notable omission given that replay is one of the most standard approaches in continual learning for reducing forgetting. Since replay is often strong in practice, its absence makes it harder to judge how much DiSC improves over a competitive conventional alternative.

- The capability-preserving checkpoint selection rule is useful, but it is also heuristic.

---

> ### Author Rebuttal · Authors · 2026-03-31
>
> Thank you for your positive assessment of our work, and for recognizing the simplicity, practical efficiency, and strong empirical performance of our approach.
>
> **W1: How does DiSC perform when combined with replay data?**
>
> We appreciate the suggestion. We evaluated replay separately when combined with regular finetuning. Counterintuitively, we found that it increased forgetting rather than mitigating it. We refer the reviewer to our response to Reviewer 8P1T for more details. As replay does not help with finetuning, we do not run additional experiments combining it with DiSC.
>
> **W2: Capability-preserving checkpoint is heuristic.**
>
> You raise a good point which we should discuss further in the paper. We note that all continual learning methods result in different trade-offs between domain adaptation and capability preservation, which makes apples-to-apples comparisons generally difficult. So, to ensure fair comparison, we report results on checkpoints selected using a transparent and interpretable criterion that one might expect to use in practice. For e.g., if we were fine-tuning on a particular dataset D, it is reasonable to expect that a practitioner would select a checkpoint that maximizes performance on D (as this is the target domain) but sets a limit on the maximum degradation on other prior capabilities.
>
> In addition to comparing the capability preserving checkpoint, we also compare DiSC’s and FT’s pareto frontiers across the full learning rate range. Figure 2 of the paper shows that DiSC’s Pareto frontier dominates FT’s. Furthermore, in Table 6 in Appendix A, we report results when selecting checkpoints using a different interpretable “heuristic”, i.e. the one that maximizes domain adaptation. We find that DiSC outperforms FT here as well; e.g., for Qwen2.5-7B-IT DiSC achieves a max KUP performance of 47.59 compared to 40.9 for FT, while still incurring substantially less forgetting.

---

> > ### Author Rebuttal · Reviewer_nj21 · 2026-04-02
> >
> > I thank the authors for providing their rebuttal. I maintain my positive score for this paper.

---

> > > ### Author Response · Authors · 2026-04-07
> > >
> > > We thank the reviewer again for their positive appraisal of our work and their continued engagement.

---

### Decision · Program_Chairs · 2026-04-30

**Decision:**

Accept (spotlight)

**Comment:**

The authors propose a simple, well motivated approach to continual adaptation of foundation models via a distillation based approach. Reviewers generally agree that the formulation of DiSC is conceptually clean, well-motivated, and practically efficient.

The empirical verification is well-executed, with strong results across Qwen-2.5 and Llama-3.1 architectures on several QA-style benchmarks. The authors also provided insightful analyses regarding domain shifts and "split design" ablations.

While the initial lack of a replay baseline was a point of discussion, the authors’ rebuttal addressed this with new experiments showing that replay was unexpectedly ineffective in preserving capabilities for this specific setup. This evidence, combined with the clear performance gains of the proposed method, ensures that the initial concern is not a blocker.

The submission is technically solid, addresses a high-impact problem, and its simplicity makes it a valuable contribution to the field.